# ADVERSARIAL WEIGHT PERTURBATION IMPROVES GENERALIZATION IN GRAPH NEURAL NETWORKS

## ABSTRACT

A lot of theoretical and empirical evidence shows that the flatter local minima tend to improve generalization. As an emerging technique to efficiently and effectively find such minima, Adversarial Weight Perturbation (AWP) minimizes the loss *w.r.t.* a bounded worst-case perturbation of the model parameters by (approximately) solving an associated min-max problem, *i.e.* favoring local minima with a small loss in a neighborhood around them. The benefits of AWP, and more generally the connections between flatness and generalization, have been extensively studied for *i.i.d.* data such as images. In this paper, we initiate the first study of this phenomenon for *non-i.i.d.* graph data. Along the way, we first derive a generalization bound for *non-i.i.d.* graph classification tasks, then we identify a vanishing-gradient issue with all existing formulations of AWP and we propose new Weighted Truncated AWP (WT-AWP) to alleviate this issue. We show that regularizing graph neural networks with WT-AWP consistently improves both natural and robust generalization across many different graph learning tasks and models.

## 1 INTRODUCTION

Simply minimizing the standard cross-entropy loss for highly non-convex and non-linear models such as (deep) neural networks is not guaranteed to obtain solutions that generalize well, especially for today's overparamatrized networks. The key underlying issue is that these models have many different local minima which can have wildly different generalization properties despite having nearly the same performance on training and validation data. Naturally, there is a rich literature that studies the properties of well-behaving local minima, as well as the design choices that improve our chances of finding them (Stutz et al., 2021). The notion of flatness which measure how quickly the loss changes in a neighbourhood around a given local minimum has been empirically shown to correlate with generalization among a variety of different measures (Jiang et al., 2019). In addition, generalization bounds based on the PAC-Bayes framework (McAllester, 1999; Foret et al., 2021) provide theoretical insights that corroborate the mounting empirical data. Since the evidence implies that flatter minima tend to generalize better, the obvious question is how to efficiently find them.

Not only do flat minima improve generalization from the training to the test data, i.e. the clean accuracy (Foret et al., 2021; Zheng et al., 2021; Kwon et al., 2021), but they also improve generalization to adversarial examples, i.e. the robust accuracy (Wu et al., 2020a; Stutz et al., 2021). Improving adversarial robustness is important, especially for models deployed in safety-critical domain or models deployed in the real-world, since most standard (undefended) models are vulnerable to adversarial attacks. Attackers can easily craft deliberate and unnoticeable input perturbations that change the prediction of the classifier. Flat minima show higher resistance to such adversarially perturbed inputs while maintaining good clean accuracy (Stutz et al., 2021).

Among the variety of techniques for finding flat minima Adversarial Weight Perturbation (AWP) (Wu et al., 2020a), and the closely-related (adaptive) sharpness-aware minimization (Foret et al., 2021; Kwon et al., 2021) and adversarial model perturbation (Zheng et al., 2021), seems to be quite effective in practice. The key idea is to minimize the loss w.r.t. a bounded worst-case perturbation of the model parameters, i.e. minimize a local notion of sharpness. The benefits of this approach, and more generally the correlation between flatness and (clean/robust) generalization, have been extensively studied for i.i.d. data such as images. In this paper we study this phenomenon for graph data for the first time. Concretely, we analyze and improve the robustness of Graph Neural Networks (GNNs) which have become a fundamental building block (in addition to CNNs and RNNs).

Blindly applying existing weight perturbation techniques to GNNs is unfortunately not effective in practice due to a vanishing-gradient issue. Intuitively, the adversarially perturbed weights tend to have a higher norm which in turn leads to a saturation in the last layer where that logits for one class are on a significantly larger scale compared to the rest. Even though this limitation plagues all formulations of AWP, for both GNNs and other models (e.g. ResNets), it has gone unnoticed so far. To address it we propose Weighted Truncated Adversarial Weight Perturbation (WT-AWP) where rather than directly minimizing the (robust) AWP loss we use it as a regularizer in addition to the standard cross-entropy loss. Moreover, we propose to abstain from perturbation in the last layer(s) of the network for a more fine-grained control of the training dynamics. These two modifications are simple, but necessary and effective. With our resulting formulation the models can obtain useful gradient signals for training even when the perturbed weights have a high norm, mitigating the gradient-vanishing issue. Furthermore, we theoretically study the AWP learning objective and show its invariance for local extrema. We can summarize our contributions as follows:

- We provide a theoretical analysis of AWP on non-i.i.d. graph data and identify a vanishing-gradient issue that plagues all previous AWP variants. Based on this analysis we propose Weighted Truncated Adversarial Weight Perturbation (WT-AWP) that mitigates this issue.

- We study for the first time the connections between flatness and generalization for Graph Neural Networks. We show that GNNs trained with our WT-AWP formulation have simultaneously improved natural and robust generalization. The improvement is statistically significant and consistent across tasks (node-level and graph-level classification) and across models (standard and robustness-aware GNNs). All this at a negligible computational cost.

## 2 BACKGROUND AND RELATED WORK

**Adversarial Weight Perturbation for Images.** AWP is motivated by the connection between the flatness of the loss landscape and model generalization. Given a learning objective $L(\cdot)$ and an image classification model with parameters $\boldsymbol{\theta}$, the generalization gap (Wu et al., 2020a), also named the sharpness term (Foret et al., 2021), which measures the worst-case flatness of the loss landscape, is defined by $[\max_{||\boldsymbol{\delta}|| \leq \rho} L(\boldsymbol{\theta} + \boldsymbol{\delta}) - L(\boldsymbol{\theta})]$. This gap is known to control a PAC-Bayes generalization bound (Neyshabur et al., 2017), with a smaller gap implying better generalization. The AWP objective optimizes the generalization gap and the loss function simultaneously via $\min_{\boldsymbol{\theta}}[(\max_{||\boldsymbol{\delta}|| \leq \rho} L(\boldsymbol{\theta} + \boldsymbol{\delta}) - L(\boldsymbol{\theta})) + L(\boldsymbol{\theta})] = \min_{\boldsymbol{\theta}} \max_{||\boldsymbol{\delta}|| \leq \rho} L(\boldsymbol{\theta} + \boldsymbol{\delta})$. Providing further theoretical justification for the effectiveness of the AWP, Zheng et al. (2021) prove that this objective favors solutions corresponding to flatter local minima assuming that the loss surface can be approximated as an inverted Gaussian surface. Relatedly, they show that AWP penalizes the gradient-norm. Keskar et al. (2016) show that large-batch training may reach sharp minima, however GNNs usually use a small batch size. In some cases we can rescale the weights to achieve arbitrarily sharp minima that also generalize well (Dinh et al., 2017). Investigating this issue for GNNs is out of our scope.

**GNNs, Graph attacks, and Graph defenses.** Graph Neural Networks (GNNs) are emerging as a fundamental building block. They have achieved spectacular results on a variety of graph learning tasks across many high-impact domains (see survey (Wu et al., 2020b)). Despite their success, it has been demonstrated that GNNs suffer from evasion attacks at test time (Zügner et al., 2018) and poisoning attacks at training time (Zügner and Günnemann, 2019). Meanwhile, a series of methods have been developed to improve their robustness. For example, GCNJaccard (Wu et al., 2019) drops dissimilar edges in the graph, as it found that attackers tend to add edges between nodes with different features. GCNSVD (Entezari et al., 2020) replaces the adjacency matrix with its low-rank approximation motivated by the observation that mostly the high frequency spectrum of the graph is affected by the adversarial perturbations. We also have provable defenses that provide robustness certificates (Bojchevski et al., 2020). Both heuristic defenses (e.g. GCNJaccard and GCNSVD) and certificates are improved with our WT-AWP. For an overview of attacks and defenses see Sun et al. (2018).

## 3 ADVERSARIAL WEIGHT PERTURBATION ON GRAPH NEURAL NETWORKS

To simplify the exposition we focus on the semi-supervised node classification task. Nonetheless, in Sec. 5.8 we show that AWP also improves graph-level classification. Let $G = (\boldsymbol{A}, \boldsymbol{X})$ be a given (attributed) graph where $\boldsymbol{A}$ is the adjacency matrix and $\boldsymbol{X}$ contains the node attributes. Let $\mathcal{V}$ be the set

of all nodes. Normally we optimize $\min_{\boldsymbol{\theta}} L_{\text{train}}(\boldsymbol{\theta}; \boldsymbol{A}, \boldsymbol{X})$, where $L_{\text{train}} = \sum_{v \in \mathcal{V}_{\text{train}}} l(f_{\boldsymbol{\theta}}(\boldsymbol{A}, \boldsymbol{X}), y_v)$, $f$ is a GNN parametrized by weights $\boldsymbol{\theta} = (\boldsymbol{\theta}_1, ..., \boldsymbol{\theta}_k)$, $y_v$ is the ground-truth label for node $v$, and $l$ is some loss function (e.g. cross-entropy) applied to each node in the training set $\mathcal{V}_{\text{train}} \subset \mathcal{V}$.

In AWP we first find the worst-case weight perturbation $\boldsymbol{\delta}^*(\boldsymbol{\theta})$ that maximizes the loss. Then we minimize the loss with the perturbed weights. The worst-case perturbation for a given $\boldsymbol{\theta}$ is defined as

$$\boldsymbol{\delta}^*(\boldsymbol{\theta}) := \arg \max_{\{\boldsymbol{\delta} \mid ||\boldsymbol{\delta}_i||_2 \leq \rho(\boldsymbol{\theta}_i), i \in [k]\}} L_{\text{train}}(\boldsymbol{\theta} + \boldsymbol{\delta}; \boldsymbol{A}, \boldsymbol{X}), \tag{1}$$

where $\rho(\boldsymbol{\theta})$ is the strength of perturbation. Then the AWP learning objective is

$$\min_{\boldsymbol{\theta}} \max_{\{\boldsymbol{\delta} \mid ||\boldsymbol{\delta}_i||_2 \leq \rho(\boldsymbol{\theta}_i), i \in [k]\}} L_{\text{train}}(\boldsymbol{\theta} + \boldsymbol{\delta}; \boldsymbol{A}, \boldsymbol{X}) = \min_{\boldsymbol{\theta}} L_{\text{train}}(\boldsymbol{\theta} + \boldsymbol{\delta}^*(\boldsymbol{\theta}); \boldsymbol{A}, \boldsymbol{X}). \tag{2}$$

Since the PAC-Bayes bound proposed by McAllester (1999) only holds for i.i.d. training tasks and semi-supervised node classification is a non-i.i.d. task, analysis in Wu et al. (2020a) and Foret et al. (2021) could not be naturally extended to node classification tasks. We derived a new generalization bound for node classification tasks on GNNs based on the sub-group generalization (Ma et al., 2021).

**Theorem 1.** *(informal). Let $L_{all}(\boldsymbol{\theta}; \boldsymbol{A}, \boldsymbol{X})$ be the loss on all nodes, including the unseen test nodes. It is bounded by the adversarially weight perturbed loss on the training nodes as follows:*

$$L_{all}(\boldsymbol{\theta}; \boldsymbol{A}, \boldsymbol{X}) \leq \max_{\{\boldsymbol{\delta} \mid ||\boldsymbol{\delta}_i||_2 \leq \rho(\boldsymbol{\theta}_i), i \in [k]\}} L_{train}(\boldsymbol{\theta} + \boldsymbol{\delta}; \boldsymbol{A}, \boldsymbol{X}) + h(||\boldsymbol{\theta}||_2^2 / \rho(\boldsymbol{\theta})^2) \tag{3}$$

The formal version, the details for $h()$, and the proof are in Appendix E. This bound justifies the use of AWP since the perturbed loss on training nodes bounds the standard loss on *all* nodes. Moreover, as $h(||\boldsymbol{\theta}||_2^2 / \rho(\boldsymbol{\theta})^2)$ is monotonously decreasing with $\rho(\boldsymbol{\theta})$, increasing the perturbation strength $\rho$ can make the bound in Eq. 3 sharper, i.e. the resulting AWP objective should lead to better generalization.

Since finding the optimal perturbation (Eq. 1) is intractable, we approximate it with a one-step projected gradient descent as in previous work (Wu et al., 2020a; Foret et al., 2021; Zheng et al., 2021),

$$\hat{\boldsymbol{\delta}}^*(\boldsymbol{\theta}) := \Pi_{B(\rho(\boldsymbol{\theta}))}(\nabla_{\boldsymbol{\theta}} L_{\text{train}}(\boldsymbol{\theta}; \boldsymbol{A}, \boldsymbol{X})), \tag{4}$$

where $B(\rho(\boldsymbol{\theta}))$ is an $l_2$ ball with radius $\rho(\boldsymbol{\theta})$ and $\Pi_{B(\rho(\boldsymbol{\theta}))}(\cdot)$ is a projection operation, which projects the perturbation back to the surface of $B(\rho(\boldsymbol{\theta}))$ when the perturbation is out of the ball. The maximum perturbation norm $\rho(\boldsymbol{\theta})$ could either be a constant (Foret et al., 2021; Zheng et al., 2021) or layer dependent (Wu et al., 2020a). We specify a layer-dependent norm constraint $\rho(\boldsymbol{\theta}) := \rho ||\boldsymbol{\theta}||_2$ because the scales of different layers in a neural network vary greatly. With the approximation $\hat{\boldsymbol{\delta}}^*(\boldsymbol{\theta})$, the definition of the final AWP learning objective is given by

$$\min_{\boldsymbol{\theta}} L_{\text{awp}}(\boldsymbol{\theta}) := L_{\text{train}}(\boldsymbol{\theta} + \Pi_{B(\rho(\boldsymbol{\theta}))}(\nabla_{\boldsymbol{\theta}} L_{\text{train}}(\boldsymbol{\theta}; \boldsymbol{A}, \boldsymbol{X})); \boldsymbol{A}, \boldsymbol{X}), \tag{5}$$

If $L_{\text{train}}(\boldsymbol{\theta}; \boldsymbol{A}, \boldsymbol{X})$ is smooth enough, $\nabla_{\boldsymbol{\theta}} L_{\text{train}}(\boldsymbol{\theta}; \boldsymbol{A}, \boldsymbol{X}) = 0$ when $\boldsymbol{\theta}^*$ is a local extremum. In this case $L_{\text{awp}}(\boldsymbol{\theta}) = L_{\text{train}}(\boldsymbol{\theta}; \boldsymbol{A}, \boldsymbol{X})$. A natural question is whether $\boldsymbol{\theta}^*$ will also be the extremum of $L_{\text{awp}}(\boldsymbol{\theta})$? We show that $L_{\text{awp}}(\boldsymbol{\theta})$ keeps the local extremum of $L_{\text{train}}(\boldsymbol{\theta}; \boldsymbol{A}, \boldsymbol{X})$ unchanged.

**Theorem 2.** *(Invariant of local minimum and maximum) With the AWP learning objective in Eq. 5, and for continuous $L_{train}(\boldsymbol{\theta}; \boldsymbol{A}, \boldsymbol{X}), \nabla_{\boldsymbol{\theta}} L_{train}(\boldsymbol{\theta}; \boldsymbol{A}, \boldsymbol{X}), \Delta_{\boldsymbol{\theta}} L_{train}(\boldsymbol{\theta}; \boldsymbol{A}, \boldsymbol{X})$, if $\boldsymbol{\theta}^*$ is a local minimum of $L_{train}(\boldsymbol{\theta}; \boldsymbol{A}, \boldsymbol{X})$ and the Hessian matrix $\Delta_{\boldsymbol{\theta}} L_{train}(\boldsymbol{\theta}; \boldsymbol{A}, \boldsymbol{X})|_{\boldsymbol{\theta}*}$ is positive definite, $\boldsymbol{\theta}^*$ is also a local minimum of $L_{awp}(\boldsymbol{\theta})$.*

The proof is provided in Appendix A. The exact gradient of this new objective is

$$\nabla_{\boldsymbol{\theta}} L_{\text{train}}(\boldsymbol{\theta} + \hat{\boldsymbol{\delta}}^*(\boldsymbol{\theta}); \boldsymbol{A}, \boldsymbol{X}) = \nabla_{\boldsymbol{\theta}} L_{\text{train}}(\boldsymbol{\theta}; \boldsymbol{A}, \boldsymbol{X})|_{\boldsymbol{\theta} + \hat{\boldsymbol{\delta}}^*(\boldsymbol{\theta})} + \nabla_{\boldsymbol{\theta}} \hat{\boldsymbol{\delta}}^*(\boldsymbol{\theta}) \nabla_{\boldsymbol{\theta}} L_{\text{train}}(\boldsymbol{\theta}; \boldsymbol{A}, \boldsymbol{X})|_{\boldsymbol{\theta} + \hat{\boldsymbol{\delta}}^*(\boldsymbol{\theta})} \quad (6)$$

Since $\nabla_{\boldsymbol{\theta}} \hat{\boldsymbol{\delta}}^*(\boldsymbol{\theta})$ includes second and higher order derivative of $\boldsymbol{\theta}$, which are computationally expensive, they are omitted during training, obtaining the following approximate gradient of the AWP loss

$$\nabla_{\boldsymbol{\theta}} L_{\text{train}}(\boldsymbol{\theta}; \boldsymbol{A}, \boldsymbol{X})|_{\boldsymbol{\theta} + \hat{\boldsymbol{\delta}}^*(\boldsymbol{\theta})} \tag{7}$$

Foret et al. (2021) show the models trained with the exact gradient (Eq. 6) have almost the same performance as model trained with the estimated first-order gradient (Eq. 7).

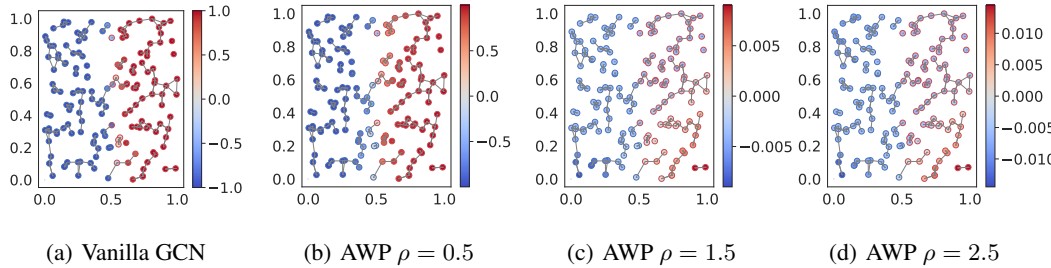

(a) Vanilla GCN      (b) AWP $\rho = 0.5$      (c) AWP $\rho = 1.5$      (d) AWP $\rho = 2.5$

Figure 1: Compare AWP models on a linearly separable dataset with different perturbation strengths $\rho$. The accuracy of models (a) to (d) is 0.97, 0.97, 0.69, and 0.48 respectively. The face color of each node shows its prediction score and the border color shows its ground-truth label. Grey lines connect the node with its nearest neighbours in the graph. For large values of $\rho$ the model is unable to learn.

## 4 WEIGHTED TRUNCATED AWP

In this section we discuss the theoretical limitations of existing AWP methods on GCN, and illustrate them empirically on a toy dataset. We also propose two approaches to improve AWP. Our improved AWP works well on both toy data and on real-world GNN benchmarks across many tasks and models. We also show that similar problems also exist for multi-layer perceptrons (see Appendix B).

### 4.1 THE VANISHING-GRADIENT ISSUE OF AWP

Consider a GCN $\hat{y} = \sigma(\hat{A}(...(\hat{A}XW_1)...)W_n)$ with a softmax activation at the output layer, where $\hat{A}$ is the graph Laplacian given by $\hat{A} := D^{-1/2}(A + I_N)D^{-1/2}, D_{ii} = \sum_j (A + I_N)_{ij}$. The perturbed model is $\hat{y} = \sigma(\hat{A}(...(\hat{A}X(W_1+\delta_1)))...(W_n+\delta_n))$. Since the norm of each perturbation $\delta_i$ could be as large as $\rho||W_i||_2$, in the worst case the norm of each layer is $(\rho + 1)||W_i||_2$, and thus the model will have exploding logit values when $\rho$ is large. After feeding large logits into the softmax layer, the output will approximate a one-hot encoded vector, because the difference between the entries of the logits may also be large. In this case the gradient will be close to 0 and the weights will not be updated. Notice, although in practice the number of GCN layers is always less than 3, we still observe the vanish gradient issue in both toy datasets and GNN benchmarks.

To verify our conclusion, we train a 2-layer GCN network with hidden dimension 64, which is a common setting for GCNs, on a linearly separable dataset. The dataset contains 2 classes $\{-1, 1\}$ and each class has 100 nodes. We apply $k$-nearest neighbor ($k = 3$) to obtain the adjacency matrix, and use the position of the nodes as the features. The number of training epochs is 200. We use 10% nodes for training, 10% for validating and the rest 80% for testing. In Fig. 1 we show the trained classifiers for different $\rho$ values. Models with AMP crash quickly when $\rho$ increased from 0.5 to 2.5. When $\rho = 0.5$, the classification accuracy is 0.97, which is nearly the same as the vanilla model, but when $\rho = 2.5$, the classification accuracy is 0.51, which is the same as a random guess. Besides, when $\rho = 1.5$ and 2.5, the loss of AWP method is almost constant dur-

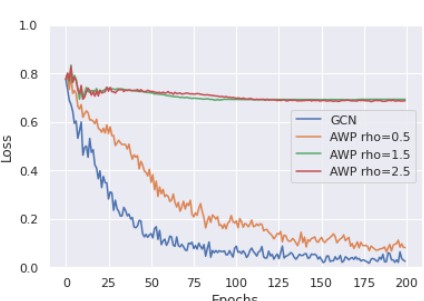

Figure 2: Learning curves for GCN and GCN+AWP with different $\rho$.

ing training (Fig. 2) and the prediction score (Fig. 1(c) and Fig. 1(d)) is around 0. This indicates that the weights are barely updated during training. So with the AWP objective, we cannot select a large $\rho$. Yet, as we discussed in Sec. 3, we prefer larger values of $\rho$ since they lead to a tighter bound (Eq. 3) and are more like to generalize better. As we shown next, our suggested improvements fix this issue.

### 4.2 TRUNCATED AWP AND WEIGHTED AWP

**Intuition for WT-AWP.** The vanishing gradient is mainly due to the exploding of the logit values, which is caused by perturbing all layers in the model. Thus, a natural idea is to only apply AWP on certain layers to mitigate the issue. This it the truncated AWP. Another idea is to provide a second

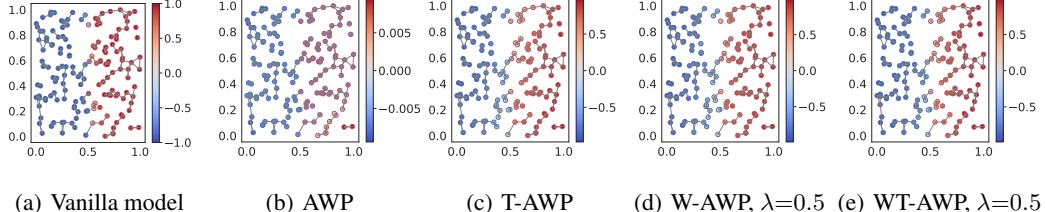

(a) Vanilla model     (b) AWP     (c) T-AWP     (d) W-AWP, $\lambda$=0.5 (e) WT-AWP, $\lambda$=0.5

Figure 3: Linearly separable dataset, $\rho = 2.5$. The accuracy of models (a) to (d) is 0.97, 0.51, 0.96, and 0.98 respectively. The face color of each node shows its prediction score and the border color shows the ground-truth label. Grey lines connect the node with its nearest neighbours in the graph.

---

**Algorithm 1** Weighted Truncated Adversarial Weight Perturbation

---

**Input:** Graph $G = (\boldsymbol{A}, \boldsymbol{X})$; model parameters $\boldsymbol{\theta} = [\boldsymbol{\theta}^{(awp)}; \boldsymbol{\theta}^{(n)}]$ with and without AWP; number of epochs $N$; loss function $L_{\text{train}}$; perturbation strength $\rho$, AWP weight $\lambda$; learning rate $\alpha$.

Initialize weight $\boldsymbol{\theta}_0$;

  **for** $t \in 1{:}N$ **do**

       Compute the loss for training nodes: $L_{\text{train}}(\boldsymbol{\theta}_{t-1}; \boldsymbol{A}, \boldsymbol{X})$

       Compute the approximating weight perturbation for $\boldsymbol{\theta}_{t-1}^{(awp)}$: $\hat{\boldsymbol{\delta}}^{*}(\boldsymbol{\theta}_{t-1}^{(awp)})$ via Eq. 4

       Compute the approximating gradient for $\boldsymbol{\theta}$:

       $g = \lambda \nabla_{\boldsymbol{\theta}} L_{\text{train}}(\boldsymbol{\theta}; \boldsymbol{A}, \boldsymbol{X})|_{\boldsymbol{\theta}_{t-1}+[\hat{\boldsymbol{\delta}}^{*}(\boldsymbol{\theta}_{t-1}^{(awp)}),0]} + (1-\lambda) \nabla_{\boldsymbol{\theta}} L_{\text{train}}(\boldsymbol{\theta}; \boldsymbol{A}, \boldsymbol{X})|_{\boldsymbol{\theta}_{t-1}}$

       Update the weight via $\boldsymbol{\theta}_t = \boldsymbol{\theta}_{t-1} - \alpha g$

**end**
**return** $\boldsymbol{\theta}_N$

---

source of valid gradients which we do by adding the the vanilla loss $L_{\text{train}}(\boldsymbol{\theta}; \boldsymbol{A}, \boldsymbol{X})$ to the AWP loss. Even when the AWP loss suffers from the vanishing gradient issue, the vanilla loss is not affected.

**Definition 1.** *(Truncated AWP) We split the model parameters into two parts* $\boldsymbol{\theta} = [\boldsymbol{\theta}^{(awp)}, \boldsymbol{\theta}^{(normal)}]$, *and we only perform AWP on* $\boldsymbol{\theta}^{(awp)}$. *The Truncated AWP objective is*

$$\min_{\boldsymbol{\theta}} L_{train}(\boldsymbol{\theta} + [\hat{\boldsymbol{\delta}}^{(awp)*}(\boldsymbol{\theta}^{(awp)}), 0]; \boldsymbol{A}, \boldsymbol{X}), \qquad (8)$$

*where* $\hat{\boldsymbol{\delta}}^{(awp)*}(\boldsymbol{\theta}^{(awp)}) := \Pi_{B(\rho(\boldsymbol{\theta}^{(awp)}))}(\nabla_{\boldsymbol{\theta}^{(awp)}} L_{train}(\boldsymbol{\theta}^{(awp)}; \boldsymbol{A}, \boldsymbol{X})).$

Recall in Sec. 2, the AWP objective is the unweighted combination of the regular loss function $L(\boldsymbol{\theta})$ and the sharpness term $\max_{\boldsymbol{\delta} \leq \rho}[L(\boldsymbol{\theta} + \boldsymbol{\delta}) - L(\boldsymbol{\theta})]$. The weight perturbation in this term can lead to vanishing gradients as we discussed in Sec. 4.1. Therefore, another way to deal with this issue is to assign a smaller weight $\lambda$ to the sharpness term in the AWP objective. The weighted combination is $[\lambda \max_{\boldsymbol{\delta} \leq \rho}[L(\boldsymbol{\theta} + \boldsymbol{\delta}) - L(\boldsymbol{\theta})] + L(\boldsymbol{\theta})] = [\lambda \max_{\boldsymbol{\delta} \leq \rho} L(\boldsymbol{\theta} + \boldsymbol{\delta}) + (1 - \lambda)L(\boldsymbol{\theta})]$.

**Definition 2.** *(Weighted AWP) Given a weight* $\lambda \in [0, 1]$ *the Weighted AWP objective is*

$$\min_{\boldsymbol{\theta}}[\lambda L_{train}(\boldsymbol{\theta} + \hat{\boldsymbol{\delta}}^{*}(\boldsymbol{\theta}); \boldsymbol{A}, \boldsymbol{X}) + (1 - \lambda)L_{train}(\boldsymbol{\theta}; \boldsymbol{A}, \boldsymbol{X})] \qquad (9)$$

We compare these two improvements with AWP and natural training on a linearly separable dataset using the same setup as in Sec. 4.1. Fig. 3 illustrates the trained models with $\rho = 2$. In Fig. 3(b) we can see that the model with AWP objective suffers from vanishing gradients and it fails to learn anything useful. The models with Truncated AWP[1] ($\boldsymbol{\theta}^{(awp)}$ = first layer and $\boldsymbol{\theta}^{(normal)}$ = last layer) (Fig. 3(c)) and Weighted AWP (Fig. 3(e)) work well and have relatively good performance (96% and 98% accuracy respectively). Besides, comparing to the vanilla model (Fig. 3(a)), they both have a significantly smoother decision boundary.

In order to obtain a more powerful approach against the vanishing-gradient issue, we combine Truncated AWP and Weighted AWP, into a Weighted Truncated Adversarial Weight Perturbation (WT-AWP). The details of WT-AWP are shown in Algorithm 1 (description in Sec. B.1). WT-AWP has two important parameters $\lambda$ and $\rho$. We will study how they influence model performance in Sec. 5.7.

---

[1]In Fig. 3(c) we perturb only the first-layer. Perturbing only the second layer instead performs similarly.

## 5    EXPERIMENTAL EVALUATIONS

We conduct comprehensive experiments to show the effect of our WT-AWP on the natural and robustness performance of different GNNs for both node classification and graph classification tasks.

**Training frameworks and setup.** We utilize the open-source libraries **Pytorch-Geometric** (Fey and Lenssen, 2019) and **Deep-Robust** (Li et al., 2020) for evaluation clean and robust node classification performance respectively. To achieve fair comparison we keep the same training settings for all models. We report the mean and standard deviation over 20 different train/val/test splits and 10 random weight initializations. See Sec. D.4 for further details and for the training hyperparameters.

**Datasets.** We use three benchmark datasets, including two citation networks, Cora and Citeseer (Sen et al., 2008), and one blog dataset Polblogs (Adamic and Glance, 2005). We treat all graphs as undirected and only select the largest connected component (more details and statistics in Sec. D.3).

**Baseline models and attacks.** We aim to evaluate the impact of our WT-AWP on natural and robust node classification tasks. We train three vanilla GNNs: GCN (Kipf and Welling, 2017), GAT (Veličković et al., 2018), and PPNP (Klicpera et al., 2018), and four graph defense methods: RGCN (Zhu et al., 2019)[2], GCNJaccard (Wu et al., 2019), GCNSVD (Entezari et al., 2020), and SimpleGCN (Jin et al., 2021). For detailed baseline descriptions see Sec. D.1. To generate the adversarial perturbations, we apply three methods including: DICE (Waniek et al., 2018), PGD (Xu et al., 2019), and Metattack (Zügner and Günnemann, 2019). For a discussion of the attacks see Sec. D.2.

**Certified robustness.** We obtain provable guarantees for our models using a black-box (sparse) randomized smoothing certificate (Bojchevski et al., 2020). We report the the certified accuracy, i.e. the percentage of nodes guaranteed to be correctly classified, given an adversary that can delete up to $r_d$ edges or add up to $r_a$ edges to graph (similarly for the node features). See Sec. D.8 for details.

**Settings for WT-AWP.** All baseline models have a 2-layer structure. When applying the WT-AWP objective, we only perform weight perturbation on the **first** layer i.e. we assign $\theta^{(awp)}$ = first layer and $\theta^{(normal)}$ = last layer. For generating the weight perturbation we use a 1-step PGD as discussed in Sec. 3. In the ablation study Sec. 5.7 we also apply 5-step PGD to generate weight perturbation, in which we utilize SGD optimizer with learning rate 0.2 and update the perturbation for 5 steps, and finally the perturbation is projected on the $l_2$ ball $B(\rho(\theta))$.

### 5.1    CLEAN ACCURACY

We evaluate the clean accuracy of node classification tasks for different GNNs and benchmarks. The baseline methods include GCN, GAT, and PPNP . We use a 2-layer structure (input-hidden-output) for these three models. For GCN and PPNP, the hidden dimensionality is 64; for GAT, we use 8 heads with size 8. We choose $K = 10, \alpha = 0.1$ in PPNP. We also find that the hyperparameters $(\lambda, \rho)$ of WT-AWP are more related to the dataset than the backbone models. We use $(\lambda = 0.7, \rho = 1)$ for all three baseline models on Cora, $(\lambda = 0.7, \rho = 2.5)$ on Citeseer, and $(\lambda = 0.3, \rho = 1)$ for GCN, $(\lambda = 0.3, \rho = 2)$ for GAT and PPNP on Polblogs. Table 1 show our results, WT-AWP clearly improves the accuracy of all baseline models, while having smaller standard deviations. Note, we do not claim that these models are state of the art, but rather that WT-AWP provides consistent and statistically significant (two-sided t-test, $p < 0.001$) improvements over the baseline models. These results support our claim that WT-AWP finds local minima with better generalization properties.

### 5.2    AVERAGE OF GRADIENT NORM IN THE INPUT SPACE

To estimate the smoothness of the loss landscape around the adjacency matrix $A$ and the node attributes $X$, we compute the average norm of the gradient of $L_{\text{train}}(\theta; A, X)$ *w.r.t.* $A$ and $X$. We compare a vanilla GCN model with GCN+WT-AWP ($\lambda = 0.5, \rho = 1$) model on Cora and Citeseer. We train 10 models with different random initializations. For each model we randomly sample 100 noisy inputs around $A$ and $X$, and we average the gradient norm for these noisy inputs. When comparing models trained with and without WT-AWP, we keep everything else fixed, including the random initialization, to isolate the effect of WT-AWP. In Fig. 4, we can observe that in most cases (37 out of 40) the models trained with WT-AWP have both better accuracy and smaller average gradient norm, *i.e.* are smoother. This provides evidence that WT-AWP can help us find flatter minima.

---

[2]Note, we cannot apply WT-AWP to RGCN as the weights are modeled by distributions.

Table 1: Clean accuracy comparison. We report the average and the standard deviation across 200 experiments per model (20 random splits × 10 random initializations). WT-AWP consistently outperform the standard models on all benchmarks. The improvements are statistically significant according to a two-sided t-test at a significance level of $p < 0.001$.

| Approachs | Cora | Citeseer | Polblogs |
|---|---|---|---|
| GCN | 84.14 ± 0.61 | 73.44 ± 1.35 | 95.04 ± 0.66 |
| GCN+WT-AWP | 85.16 ± 0.44 | 74.48 ± 1.04 | 95.26 ± 0.51 |
| GAT | 84.13 ± 0.79 | 73.71 ± 1.23 | 94.93 ± 0.51 |
| GAT+WT-AWP | 85.13 ± 0.51 | 74.73 ± 1.07 | 95.12 ± 0.48 |
| PPNP | 85.56 ± 0.46 | 74.50 ± 1.06 | 95.18 ± 0.42 |
| PPNP+WT-AWP | **86.13 ± 0.43** | **75.64 ± 0.95** | **95.36 ± 0.37** |

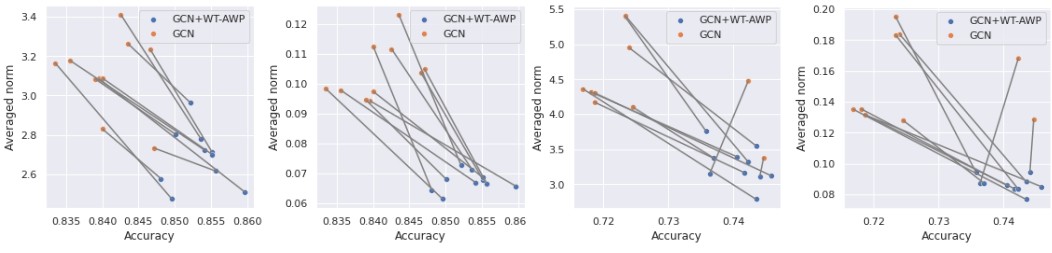

(a) Cora adj. matrix     (b) Cora node feat.     (c) Citeseer adj. matrix     (d) Citeseer node feat.

Figure 4: Comparison of the averaged gradient norm w.r.t. the adjacency matrix and the node features for GCN models with and without WT-AWP on Cora and Citeseer. Each connected pair of points refers to a GCN and a GCN+WT-AWP model trained with the same data split and initialization.

## 5.3 VISUALIZATION OF LOSS LANDSCAPE

We train GCN, GCN+AWP ($\rho = 0.1$) and GCN+WT-AWP ($\lambda = 0.5, \rho = 0.5$) models with the same initialization, and we compare their loss landscapes. The accuracy is 83.55% for GCN, 84.21% for GCN+AWP, and 85.51% for GCN+WT-AWP. Similar to Stutz et al. (2021), Fig. 5 shows the loss landscape in a randomly chosen direction $u$ in weight space, *i.e.* we plot $L_{\text{train}}(\theta + \alpha \cdot u; A, X)$ for different steps $\alpha$. We generate 10 random directions $u$ and show the average loss. The loss landscape of GCN+AWP is slightly smoother than the vanilla GCN, because of the small perturbation strength $\rho = 0.1$.

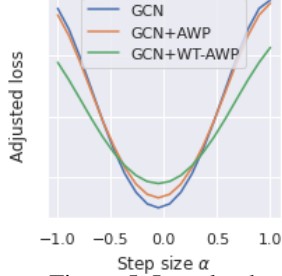

Figure 5: Loss landscape.

GCN+WT-AWP is flatter (and more accurate) than both of them due to the larger perturbation strength $\rho = 0.5$. This provides further evidence for the effectiveness of WT-AWP.

## 5.4 ROBUST ACCURACY WITH POISONING ATTACKS

Next we show that our WT-AWP can improve existing defense methods against graph poisoning attacks. We select two poisoning attacks: PGD and Metattack (Zügner and Günnemann, 2019), with a 5% adversarial budget. The baseline models are vanilla GCN, and three GCN-based graph-defense models: GCNJaccard, GCNSVD, and SimpleGCN. For all attack and defense methods, we apply the default hyperparameter settings in Li et al. (2020), which re-implements the corresponding models with the same hyperparameters as the original works. We use Cora, Citeseer, and Polblogs as the benchmark datasets. Note that GCNJaccard does not work on Polblogs as it requires node features. Table 10 in the appendix shows the hyperparameters $(\lambda, \rho)$ we select for all WT-AWP models.

As we can see in Table 2, none of the defense methods have dominant performance across benchmarks. More importantly, our WT-AWP consistently improves the robust accuracy for both vanilla and robust models. We also evaluate the models against the DICE poisoning attack in Sec. C.2, and again the results demonstrate that WT-AWP adds meaningful improvement over the baselines.

Table 2: Robust accuracy under PGD and Metattack poisoning attacks, with a 5% adversarial budget. We report the average and the standard deviation across 200 experiments per model (20 random splits × 10 random initializations). Our WT-AWP loss improves over all (vanilla and robust) baselines. All results expect the one marked with * are statistically significant at $p < 0.05$ according to a t-test.

| | Natural Acc | | | Acc with 5% PGDattack | | | Acc with 5% Metattack | | |
|---|---|---|---|---|---|---|---|---|---|
| Models | Cora | Citeseer | Polblogs | Cora | Citeseer | Polblogs* | Cora | Citeseer | Polblogs |
| GCN | 83.73 ± 0.71 | 73.03 ± 1.19 | 95.06 ± 0.68 | 81.26 ± 1.27 | 72.04 ± 1.60 | 85.18 ± 2.63 | 78.61 ± 1.66 | 69.20 ± 1.93 | 79.74 ± 1.05 |
| +WT-AWP | **84.66 ± 0.53** | 74.01 ± 1.11 | **95.20 ± 0.61** | 82.66 ± 1.07 | 73.73 ± 1.23 | **85.73 ± 4.17** | 79.05 ± 1.73 | 70.50 ± 1.65 | 80.72 ± 1.25 |
| GCNJaccard | 82.42 ± 0.73 | 73.09 ± 1.20 | N/A | 80.65 ± 1.14 | 72.05 ± 1.76 | N/A | 78.96 ± 1.54 | 69.62 ± 1.87 | N/A |
| +WT-AWP | 83.55 ± 0.60 | 74.10 ± 1.04 | N/A | 82.12 ± 0.91 | 73.85 ± 1.38 | N/A | **80.23 ± 1.38** | 71.22 ± 1.44 | N/A |
| SimPGCN | 82.99 ± 0.68 | 74.05 ± 1.28 | 94.67 ± 0.95 | 80.71 ± 1.33 | 73.61 ± 1.39 | 82.42 ± 3.14 | 78.60 ± 1.81 | 72.52 ± 1.72 | 76.66 ± 1.80 |
| +WT-AWP | 83.37 ± 0.74 | **74.26 ± 1.09** | 94.85 ± 0.91 | **83.49 ± 0.78** | **74.43 ± 1.14** | 82.68 ± 4.82 | 79.76 ± 1.76 | **72.95 ± 1.43** | 77.68 ± 2.41 |
| GCNSVD | 77.63 ± 0.63 | 68.57 ± 1.54 | 94.08 ± 0.59 | 76.83 ± 1.42 | 68.08 ± 1.98 | 82.84 ± 3.05 | 76.28 ± 1.15 | 67.34 ± 1.93 | 91.76 ± 1.19 |
| +WT-AWP | 79.05 ± 0.58 | 71.12 ± 1.42 | 94.13 ± 0.59 | 78.50 ± 0.89 | 71.43 ± 1.46 | 82.97 ± 3.57 | 77.61 ± 1.08 | 70.65 ± 1.28 | **92.28 ± 0.98** |
| RGCN | 83.29 ± 0.63 | 71.69 ± 1.35 | 95.15 ± 0.46 | 78.47 ± 1.10 | 68.81 ± 2.32 | 85.62 ± 1.51 | 77.70 ± 1.69 | 69.05 ± 1.90 | 79.48 ± 1.16 |

Table 3: Robust accuracy under evasion attacks of different strength. We report the average and the standard deviation across 200 experiments per model (20 random splits × 10 random initializations). Our WT-AWP loss always improves the robustness of the baseline models.

| | Perturbation strength | 5% | | | 10% | | |
|---|---|---|---|---|---|---|---|
| Attacks | Models | Cora | Citeseer | Polblogs | Cora | Citeseer | Polblogs |
| DICE | GCN | 82.83 ± 0.87 | 71.85 ± 1.31 | 91.27 ± 0.98 | 81.87 ± 0.94 | 71.17 ± 1.50 | 87.47 ± 1.17 |
| | +WT-AWP | **84.01 ± 0.59** | **73.84 ± 1.10** | **91.45 ± 0.86** | **82.93 ± 0.64** | **73.14 ± 1.25** | **87.70 ± 0.97** |
| PGD | GCN | 79.92 ± 0.62 | 70.50 ± 1.35 | 79.41 ± 0.76 | 77.17 ± 0.74 | 68.49 ± 1.39 | 72.90 ± 0.73 |
| | +WT-AWP | **81.00 ± 0.56** | **70.69 ± 1.45** | **80.70 ± 0.90** | **77.87 ± 0.64** | **68.96 ± 1.30** | **75.11 ± 1.03** |

## 5.5 ROBUST ACCURACY WITH EVASION ATTACKS

Next we show that WT-AWP also improves existing defense methods against graph evasion attacks. We select two evasion attacks, DICE and PGD, with perturbation strengths of 5% and 10%. The baseline model is GCN and we perform experiments on three benchmarks: Cora, Citeseer, and Polblogs. For the PGD attack the hyperparameters $(\lambda, \rho)$ are (0.5, 0.5) for all datasets. For the DICE attack we use (0.5, 0.5) for Cora, (0.7, 2) for Citeseer, and (0.3, 1) for Polblogs. Table 3 shows the experimental results. WT-AWP again meaningfully improves the robustness of GCN under both PGD and DICE evasion attacks.

## 5.6 CERTIFIED ROBUSTNESS

In this subsection, we measure the certified robustness of GCN and GCN+WT-AWP on the Cora dataset with sparse randomized smoothing (Bojchevski et al., 2020). We use $\lambda = 0.5, \rho = 1$ as the hyperparameters for the WT-AWP models. We plot the certified accuracy $S(r_a, r_d)$ for different addition $r_a$ and deletion $r_d$ radii. In Fig. 6, we see that compared to vanilla GCN training, our WT-AWP loss significantly increases the certified accuracy *w.r.t.* feature perturbations for all radii, while maintaining comparable performance when certifying perturbations of the graph structure. For additional results see Sec. C.3.

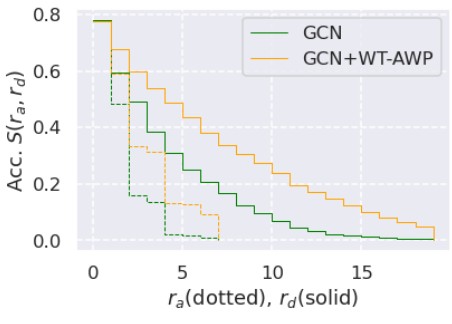
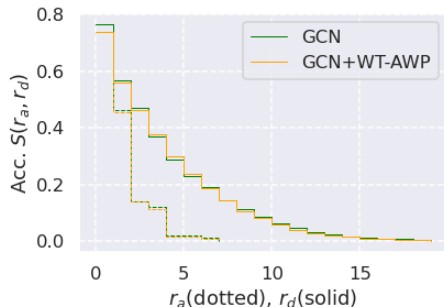

(a) Node feature perturbations          (b) Graph structure perturbations

Figure 6: Robustness guarantees on Cora. WT-AWP improves the certificate for node features.

Table 4: Hyperparameter sensitivity study for $\lambda$ and $\rho$ on the Cora dataset for a GCN base model.

| WT-AWP | $\rho = 0.05$ | $\rho = 0.1$ | $\rho = 0.5$ | $\rho = 1$ | $\rho = 2.5$ | $\rho = 5$ |
|---|---|---|---|---|---|---|
| $\lambda = 0.1$ | $84.15 \pm 0.60$ | $84.15 \pm 0.61$ | $84.51 \pm 0.48$ | $84.58 \pm 0.52$ | $84.50 \pm 0.51$ | $84.54 \pm 0.49$ |
| $\lambda = 0.3$ | $84.10 \pm 0.62$ | $84.13 \pm 0.58$ | $84.76 \pm 0.51$ | $84.91 \pm 0.46$ | $84.77 \pm 0.46$ | $84.64 \pm 0.47$ |
| $\lambda = 0.5$ | $84.11 \pm 0.64$ | $84.09 \pm 0.61$ | $84.93 \pm 0.49$ | $85.06 \pm 0.49$ | $84.94 \pm 0.45$ | $84.67 \pm 0.49$ |
| $\lambda = 0.7$ | $84.13 \pm 0.59$ | $84.15 \pm 0.64$ | $85.00 \pm 0.46$ | $\mathbf{85.16 \pm 0.44}$ | $84.99 \pm 0.49$ | $84.66 \pm 0.49$ |
| $\lambda = 1.0$ | $84.12 \pm 0.69$ | $84.23 \pm 0.64$ | $82.45 \pm 1.98$ | $60.29 \pm 1.94$ | $29.51 \pm 0.91$ | $29.19 \pm 0.13$ |
| AWP | $84.16 \pm 0.68$ | $84.23 \pm 0.68$ | $41.19 \pm 1.23$ | $29.18 \pm 0.07$ | $29.18 \pm 0.02$ | $29.18 \pm 0.02$ |
| W-AWP | $84.12 \pm 0.66$ | $84.20 \pm 0.66$ | $84.63 \pm 0.51$ | $84.32 \pm 0.65$ | $83.98 \pm 0.93$ | $83.62 \pm 1.27$ |

Table 5: Ablation study with $\lambda$ and $\rho$ on WT-AWP, where the weight perturbation is calculated with 5-step PGD. The backbone model is GCN and the benchmark is Cora.

| WT-AWP (5 step) | $\rho = 0.05$ | $\rho = 0.1$ | $\rho = 0.5$ | $\rho = 1$ | $\rho = 2.5$ | $\rho = 5$ |
|---|---|---|---|---|---|---|
| $\lambda = 0.1$ | $84.19 \pm 0.60$ | $84.17 \pm 0.59$ | $84.45 \pm 0.51$ | $84.50 \pm 0.50$ | $84.39 \pm 0.52$ | $84.41 \pm 0.54$ |
| $\lambda = 0.3$ | $84.12 \pm 0.58$ | $84.15 \pm 0.63$ | $84.65 \pm 0.54$ | $84.81 \pm 0.47$ | $84.70 \pm 0.50$ | $84.55 \pm 0.55$ |
| $\lambda = 0.5$ | $84.10 \pm 0.59$ | $84.11 \pm 0.62$ | $84.77 \pm 0.53$ | $84.90 \pm 0.50$ | $84.82 \pm 0.47$ | $84.64 \pm 0.52$ |
| $\lambda = 0.7$ | $84.12 \pm 0.61$ | $84.11 \pm 0.63$ | $84.86 \pm 0.49$ | $\mathbf{84.99 \pm 0.48}$ | $84.89 \pm 0.51$ | $84.64 \pm 0.52$ |
| $\lambda = 1.0$ | $84.11 \pm 0.62$ | $84.18 \pm 0.63$ | $72.18 \pm 1.48$ | $32.55 \pm 6.80$ | $29.18 \pm 0.03$ | $29.18 \pm 0.00$ |

## 5.7 Ablation and Hyperparameter Sensitivity Study

We compare the performance of GCN+WT-AWP on the Cora dataset for different $\lambda$ and $\rho$ values. We also compare GCN+WT-AWP with GCN+AWP under different perturbation sizes $\rho$. Table 4 lists the results. The accuracy of GCN+WT-AWP first increases with $\lambda$ and $\rho$ and then slightly decreases. Truncated AWP is a special case for $\lambda = 1$ (since the $(1-\lambda)$ term disappears in Eq. 9) and it does not perform well, especially for larger $\rho$. Similarly, WT-AWP outperforms the vanilla AWP that suffers from the vanishing-gradient issue. Weighted but not truncated AWP with $\lambda = 0.5$ (last row) is also worse than WT-AWP, although in general weighting seems to be more important than truncation. These results justify the decision to combine our proposed weighted and truncated AWP methods.

We also generate perturbations as in Eq. 4 but with multi-step PGD. We repeat the above experiment with applying 5-step PGD for WT-AWP. As shown in Table 5, the performance of 5-step WT-AWP is similar to the 1-step WT-AWP, the accuracy of both models first increases with $\lambda$ and $\rho$, and then decreases. The optimal hyperparameters $(\lambda, \rho)$ are $\rho = 1, \lambda = 0.7$. Since 5-step PGD offers no benefits and 1-step PGD is faster, we suggest this as the default setting when applying WT-AWP.

## 5.8 Graph Classification

Finally, we conduct experiments on graph classification tasks with three benchmark datasets: Protein, IMDB-Binary and IMDB-Multi. Detailed description is in Sec. D.9. Table 6 shows the experimental results. Generally, WT-AWP improves the accuracy with a large margin. Besides, the variance of the accuracy of GCN+WT-AWP across different random seeds is significantly smaller than the vanilla GCN, which indicates that WT-AWP is also more stable. Note, we do not claim that our models are state of the art, but rather that WT-AWP provides consistent improvements.

Table 6: Performance of WT-AWP on graph classification tasks, the backbone is GCN.

| | Proteins | IMDB-Binary | IMDB-Multi |
|---|---|---|---|
| GCN | $75.05 \pm 1.40$ | $72.40 \pm 2.73$ | $55.53 \pm 1.33$ |
| GCN+WT-AWP | $\mathbf{76.48 \pm 0.49}$ | $\mathbf{75.80 \pm 1.17}$ | $\mathbf{57.26 \pm 0.63}$ |

## 6 Conclusion

We proposed a new adversarial weight perturbation method, WT-AWP, and we evaluated it on graph neural networks. We showed that our WT-AWP can improve the regularization of GNNs by finding flat local minima. We conducted extensive experiments to validate our method. In all empirical results, WT-AWP consistently improves the performance of GNNs on a wide range of graph learning tasks including node classification, graph defense, and graph classification. Further exploring the connections between flat minima and generalization in GNNs is a promising research direction.

REPRODUCIBILITY STATEMENT

All datasets, baseline models, and general training settings are listed at the beginning of Sec. 5. For specific tasks we include the detailed settings in the corresponding sections. For example, the detailed model structure and hyperparameter settings for GCN clean accuracy is discussed in Sec. 5.1. We will make our code available to the reviewers via an anonymous link posted on OpenReview as suggested by the guidelines.

ETHICS STATEMENT

In this paper we design a new regularization method that can improve the robustness of graph neural networks again adversarial attacks. Making GNNs more robust can have positive or negative broader impacts depending on the application and the domain. While we observed that WT-AWP improves both clean and robust generalization, we did not study whether these improvements come at a cost to e.g. the fairness of the model, or whether they introduce certain biases in the model.

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

# Supplementary Material

## A  PROOFS

*Proof.* (Theorem 2) We only show the proof with local minimums, the proof with local maximum is analogous. For ease of calculation we denote $L_{\text{train}}(\boldsymbol{\theta}; \boldsymbol{A}, \boldsymbol{X})$ by $L(\boldsymbol{\theta})$. We need to show a) $\nabla_{\boldsymbol{\theta}} L(\boldsymbol{\theta} + \nabla_{\boldsymbol{\theta}} L(\boldsymbol{\theta}))|_{\boldsymbol{\theta}*} = 0$, and b) $\Delta_{\boldsymbol{\theta}} L(\boldsymbol{\theta} + \nabla_{\boldsymbol{\theta}} L(\boldsymbol{\theta}))|_{\boldsymbol{\theta}*}$ is positive definite.

a) Since $\boldsymbol{\theta}^*$ is a local minimum of $L$, we have $\nabla_{\boldsymbol{\theta}} L(\boldsymbol{\theta})|_{\boldsymbol{\theta}^*} = 0$, thus

$$
\begin{aligned}
\nabla_{\boldsymbol{\theta}} L(\boldsymbol{\theta} + \nabla_{\boldsymbol{\theta}} L(\boldsymbol{\theta}))|_{\boldsymbol{\theta}*} &= (I + \Delta_{\boldsymbol{\theta}} L(\boldsymbol{\theta}^*)) \nabla_{\boldsymbol{\theta}} L(\boldsymbol{\theta})|_{\boldsymbol{\theta}* + \nabla_{\boldsymbol{\theta}} L(\boldsymbol{\theta})|_{\boldsymbol{\theta}*}} \\
&= (I + \Delta_{\boldsymbol{\theta}} L(\boldsymbol{\theta}^*)) \nabla_{\boldsymbol{\theta}} L(\boldsymbol{\theta})|_{\boldsymbol{\theta}*} = 0
\end{aligned}
\tag{10}
$$

b)

$$
\begin{aligned}
\nabla_{\boldsymbol{\theta}} (\nabla_{\boldsymbol{\theta}} L(\boldsymbol{\theta} + \nabla_{\boldsymbol{\theta}} L(\boldsymbol{\theta})))|_{\boldsymbol{\theta}*} &= \nabla_{\boldsymbol{\theta}} [(I + \Delta_{\boldsymbol{\theta}} L(\boldsymbol{\theta})) \nabla_{\boldsymbol{\theta}} L(\boldsymbol{\theta} + \nabla_{\boldsymbol{\theta}} L(\boldsymbol{\theta}))]|_{\boldsymbol{\theta}*} \\
&= \nabla_{\boldsymbol{\theta}} (I + \Delta_{\boldsymbol{\theta}} L(\boldsymbol{\theta}))|_{\boldsymbol{\theta}*} \nabla_{\boldsymbol{\theta}} L(\boldsymbol{\theta})|_{\boldsymbol{\theta}* + \nabla_{\boldsymbol{\theta}} L(\boldsymbol{\theta})|_{\boldsymbol{\theta}*}} \\
&\quad + (I + \Delta_{\boldsymbol{\theta}} L(\boldsymbol{\theta}))|_{\boldsymbol{\theta}*} \Delta_{\boldsymbol{\theta}} L(\boldsymbol{\theta})|_{\boldsymbol{\theta}* + \nabla_{\boldsymbol{\theta}} L(\boldsymbol{\theta})|_{\boldsymbol{\theta}*}} (I + \Delta_{\boldsymbol{\theta}} L(\boldsymbol{\theta}))^T|_{\boldsymbol{\theta}*} \\
&= (I + \Delta_{\boldsymbol{\theta}} L(\boldsymbol{\theta}))|_{\boldsymbol{\theta}*} \Delta_{\boldsymbol{\theta}} L(\boldsymbol{\theta})|_{\boldsymbol{\theta}*} (I + \Delta_{\boldsymbol{\theta}} L(\boldsymbol{\theta}))^T|_{\boldsymbol{\theta}*}
\end{aligned}
\tag{11}
$$

Because $(I + \Delta_{\boldsymbol{\theta}} L(\boldsymbol{\theta}))|_{\boldsymbol{\theta}*}$ and $\Delta_{\boldsymbol{\theta}} L(\boldsymbol{\theta})|_{\boldsymbol{\theta}*}$ are positive definite matrices, and $\nabla_{\boldsymbol{\theta}}(\nabla_{\boldsymbol{\theta}} L(\boldsymbol{\theta} + \nabla_{\boldsymbol{\theta}} L(\boldsymbol{\theta})))|_{\boldsymbol{\theta}*} = (I + \Delta_{\boldsymbol{\theta}} L(\boldsymbol{\theta}))|_{\boldsymbol{\theta}*} \Delta_{\boldsymbol{\theta}} L(\boldsymbol{\theta})|_{\boldsymbol{\theta}*} (I + \Delta_{\boldsymbol{\theta}} L(\boldsymbol{\theta}))^T|_{\boldsymbol{\theta}*}$ is symmetric, $\nabla_{\boldsymbol{\theta}}(\nabla_{\boldsymbol{\theta}} L(\boldsymbol{\theta} + \nabla_{\boldsymbol{\theta}} L(\boldsymbol{\theta})))|_{\boldsymbol{\theta}*}$ is positive definite. Thus $\boldsymbol{\theta}^*$ is also the local minimum of $L(\boldsymbol{\theta} + \nabla_{\boldsymbol{\theta}} L(\boldsymbol{\theta}))$.

$\square$

## B  VANISHING-GRADIENT ISSUE OF AWP ON MLP

In this section we show that the vanishing-gradient issue also happens in multi-layer perceptrons. Consider an MLP $\hat{y} = \sigma(\boldsymbol{W}_n(...(\boldsymbol{W}_1 \boldsymbol{X})))$ with a softmax activation at the output layer. The perturbed model is $\hat{y} = \sigma(((\boldsymbol{W}_n + \boldsymbol{\delta}_n)(...((\boldsymbol{W}_1 + \boldsymbol{\delta}_1) \boldsymbol{X})))$. Since the norm of each perturbation $\boldsymbol{\delta}_i$ could be as large as $\rho ||\boldsymbol{W}_i||_2$, in the worst case the norm of each layer is $(\rho + 1) ||\boldsymbol{W}_i||_2$, and thus the model will have exploding logit values when $\rho$ is large. After feeding large logits into the softmax layer, the output will approximate a one-hot encoded vector, because the difference between the entries of the logits will also be large. The gradient will be close to $0$ and the weights will not be updated.

To verify our conclusion we train a 3-layer linear network with $\boldsymbol{W}_1 \in \mathbb{R}^{2 \times 100}, \boldsymbol{W}_2 \in \mathbb{R}^{100 \times 100}, \boldsymbol{W}_3 \in \mathbb{R}^{100 \times 2}$ on a linearly separable dataset, the number of training epochs is 2000. In Fig. 7 we show the trained classifiers with different $\rho$. We find that models with AWP are crushed quickly when $\rho$ increased from 0.2 to 0.25. When $\rho = 0.25$, the value of loss function remains unchanged during training and the prediction score is around 0, which indicates the weights are almost not updated during training. So with the AWP objective, we cannot select a large $\rho$.

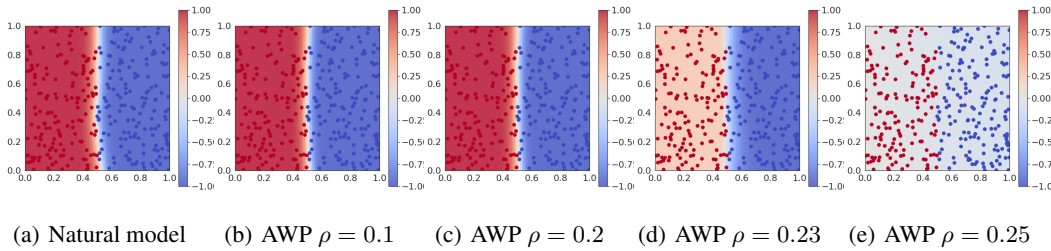

(a) Natural model    (b) AWP $\rho = 0.1$    (c) AWP $\rho = 0.2$    (d) AWP $\rho = 0.23$    (e) AWP $\rho = 0.25$

Figure 7: Comparing AWP models on a linearly separable dataset with different $\rho$ values.

We repeat the experiment using the same 3-layer linear network on a 2d moons dataset and perform the weight perturbation only on the first two layers. Fig. 8 Illustrates the trained models with $\rho = 0.4$.

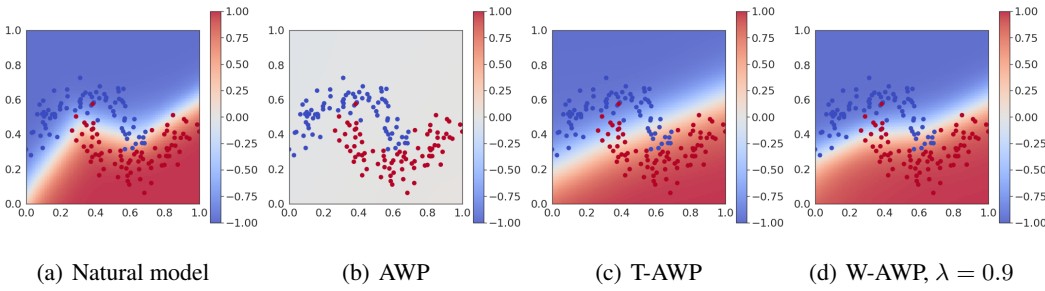

(a) Natural model     (b) AWP     (c) T-AWP     (d) W-AWP, $\lambda = 0.9$

Figure 8: Model comparison on the two moons dataset, $\rho = 0.4$

In Fig. 8(b) we can see the model suffers from vanishing gradients, while the model with truncated AWP works well (Fig. 8(c)). Besides, comparing to the overfitted natural model (Fig. 8(a)), the truncated AWP model has a smoother decision boundary. We also remove the weight perturbation on the middle or the first layer and train the model correspondingly, the results are similar to Fig. 8(c). As Fig. 8(d) shows, the model with the weighted AWP objective is able to learn the representation of the input data, and the decision boundary is also smoother than the nature model in Fig. 8(a).

### B.1 DESCRIPTION OF ALGORITHM 1

In the WT-AWP algorithm (Algorithm 1) we apply a numerical optimizer such as Adam to the WT-AWP objective

$$L_{WT-AWP}(\theta) = [\lambda L_{\text{train}}(\theta + [\hat{\delta}^{(awp)*}(\theta^{(awp)}), 0]; A, X) + (1 - \lambda)L_{\text{train}}(\theta; A, X)]$$

Since in our empirical experiments the GNNs always have a 2-layer structure, we assign $\theta^{awp}$ as the first layer and $\theta^{normal}$ as the last layer. Then the perturbation $\hat{\delta}^{(awp)*}$ (first layer) is computed via Eq. (4). Next the gradient $g$ of $L_{WT-AWP}$ is calculated with Eq. (6).

$$g = \lambda \nabla_\theta L_{\text{train}}(\theta; A, X)|_{\theta_{t-1}+[\hat{\delta}^*(\theta_{t-1}^{(awp)}),0]} + (1 - \lambda)\nabla_\theta L_{\text{train}}(\theta; A, X)|_{\theta_{t-1}}$$

Finally we update the weight via $\theta_t = \theta_{t-1} - \alpha g$.

## C ADDITIONAL EXPERIMENTS

### C.1 LEARNING CURVES AND GENERALIZATION GAP DURING TRAINING

In this part, we train GCN, GCN+AWP ($\rho = 0.1$) and GCN+WT-AWP ($\lambda = 0.5, \rho = 0.5$) models with the same random initialization and compare their learning curves, and generalization gap. The accuracy is 0.8355 for GCN, 0.8421 for GCN+AWP, and 0.8551 for GCN+WT-AWP. Fig. 9(a) illustrates the learning curve of vanilla loss $L_{\text{train}}(\theta; A, X)$ during training. The loss of all three models converges well. The final value of GCN+WT-AWP is larger than the rest two models. We believe it is because GCN+WT-AWP finds a different (flatter) local minimum. Fig. 9(b) shows the generalization gap during training. Because we use a large perturbation bound $\rho$ in GCN+WT-AWP, its generalization gap fluctuates more and decrease slower compared to the gap of GCN+AWP. The fluctuation is due to the exploding logit problem in AWP with a large $\rho$ value. When it happens, the regular loss included in WT-AWP can minimize (but not completely eliminate) its influence. Despite the fluctuation, the generalization gap of GCN+WT-AWP decreases with time as well.

### C.2 ROBUST ACCURACY WITH POISONING DICE ATTACK

We conduct additional experiments on poisoning the graph with DICE attacks. The general model settings are the same as Sec. 5.4. The WT-AWP hyperparameters $(\lambda, \rho)$ are shown in Table 8. Table 7 illustrates the experimental results. The models that achieve best performance on a given dataset are all based on WT-AWP. Besides, WT-AWP also consistently boost the performance of the baselines.

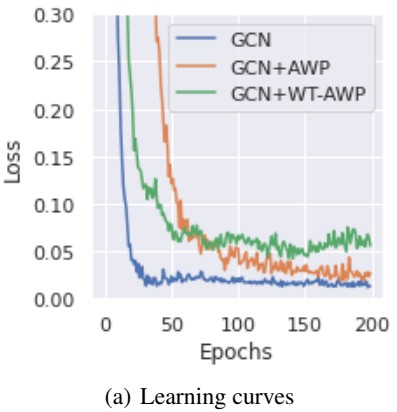
(a) Learning curves

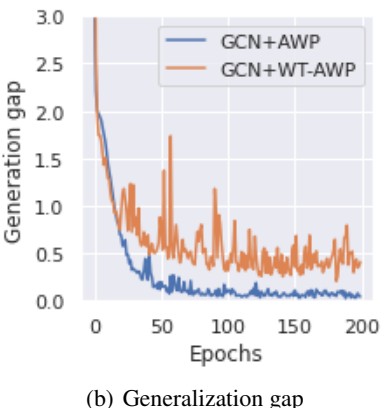
(b) Generalization gap

Figure 9: Learning Curves and Generalization Gap During Training.

Table 7: Robust accuracy with 5% poisoning DICE attacks. We report the average and the standard deviation across 200 experiments per model (20 random splits × 10 random initializations).

| | Natural Acc. | | | Acc. with 5% DICE attack | | |
|---|---|---|---|---|---|---|
| Approachs | Cora | Citeseer | Polblogs | Cora | Citeseer | Polblogs |
| GCN | 83.73 ± 0.71 | 73.03 ± 1.19 | 95.06 ± 0.68 | 82.60 ± 0.76 | 71.89 ± 1.17 | 90.13 ± 0.82 |
| +WT-AWP | **84.66 ± 0.53** | 74.01 ± 1.11 | **95.20 ± 0.61** | **83.87 ± 0.62** | 73.68 ±1.06 | 90.31 ± 0.79 |
| GCNJaccard | 82.42 ± 0.73 | 73.09 ± 1.20 | N/A | 81.55 ± 0.86 | 72.22 ± 1.22 | N/A |
| +WT-AWP | 83.55 ± 0.60 | 74.10 ± 1.04 | N/A | 82.86 ± 0.73 | **73.95 ± 1.04** | N/A |
| SimPGCN | 82.99 ± 0.68 | 74.05 ± 1.28 | 94.67 ± 0.95 | 82.11 ± 0.70 | 73.53 ± 1.23 | 89.57 ± 1.06 |
| +WT-AWP | 83.37 ± 0.74 | **74.26 ± 1.09** | 94.85 ± 0.91 | 83.30 ± 0.73 | 73.89 ± 1.08 | 90.13 ± 1.03 |
| GCNSVD | 77.63 ± 0.63 | 68.57 ± 1.54 | 94.08 ± 0.59 | 76.25 ± 0.91 | 67.27 ± 1.67 | 90.80 ± 0.88 |
| +WT-AWP | 79.05 ± 0.58 | 71.12 ± 1.42 | 94.13 ± 0.59 | 77.51 ± 0.77 | 70.30 ± 1.22 | **91.11 ± 0.76** |
| RGCN | 83.29 ± 0.63 | 71.69 ± 1.35 | 95.15 ± 0.46 | 82.02 ± 0.73 | 70.18 ± 1.38 | 90.03 ± 0.67 |

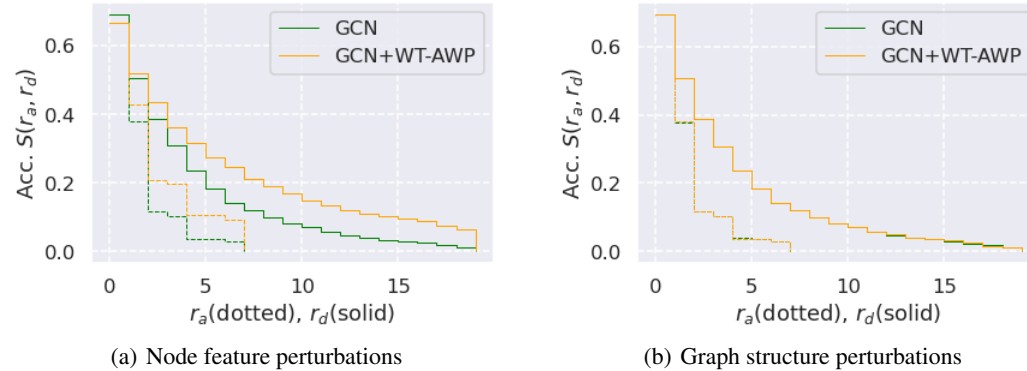

(a) Node feature perturbations          (b) Graph structure perturbations

Figure 10: Certified adversarial robustness on the Citeseer dataset.

Table 8: Hyperparameters of WT-AWP for poisoning DICE attacks

| $(\lambda, \rho)$ | Cora | Citeseer | Polblogs |
|---|---|---|---|
| GCN | | | |
| GCNJaccard | (0.5, 0.5) | (0.7, 2) | (0.3, 1) |
| GCNSVD | | | |
| SimPGCN | (0.1, 0.5) | (0.5, 0.1) | (0.5, 1) |

### C.3 CERTIFIED ROBUSTNESS ON CITESEER DATASET

We measure the certified robustness of GCN and GCN+WT-AWP with randomized smoothing (Bojchevski et al., 2020) on the Citeseer dataset. We use $\lambda = 0.5, \rho = 1$ as the hyperparameters for WT-AWP models. We plot the certified accuracy $S(r_a, r_d)$ w.r.t. $r_a$ and $r_d$. As seen in Fig. 10, comparing with the vanilla GCN, WT-AWP significantly increases the certified accuracy for perturbations to the node features for all radii, while having comparable performance for certification of the graph structure.

## D  EXPERIMENTAL DETAILS

### D.1  DESCRIPTION OF BASELINE MODELS

We aim to evaluate the impact of our WT-AWP on natural and robust node classification tasks, thus we utilize the well-known graph neural networks and graph defense methods as baseline. We first train the baseline models and compare their performance with the baseline models trained with WT-AWP objective (if applicable). The baseline GNN models include:

- **GCN** (Kipf and Welling, 2017): is one of the most representative graph convolution neural networks. Currently it can still achieve SOTA on different graph learning tasks.

- **GAT** (Veličković et al., 2018): utilizes multi-head attention mechanism to learn different weights for each node and its neighbor node without requiring the spectral decomposition.

- **PPNP** (Klicpera et al., 2018): improves the GCN propagation scheme based on the personalized Pagerank. This approach generates predictions from each node's own features and propagates these predictions using an adaptation of personalized PageRank.

- **RGCN** (Zhu et al., 2019): applies the Gaussian distribution to model the node representations. This structure is expected to absorb effects of adversarial attacks. It also penalizes nodes with large variance with an attention mechanism. Notice, the WT-AWP cannot be applied to RGCN, as the weights of RGCN are modeled by distributions. We can regard RGCN as another model inspired by PAC-Bayes theorem, as it models the objective $\mathbb{E}_{\boldsymbol{\delta} \sim \mathcal{N}(0, \sigma \boldsymbol{I})}[L_{\text{train}}(\boldsymbol{\theta} + \boldsymbol{\delta}; \boldsymbol{A}, \boldsymbol{X})]$, which also bounded $L_{\text{all}}(\boldsymbol{\theta}; \boldsymbol{A}, \boldsymbol{X})$ according to PAC-Bayes theorem.

- **GCNJaccard** (Wu et al., 2019): is a graph defense method based on GCN. It pre-processes the graph by deleting edges, which connect nodes with a small Jaccard similarity of features, because attackers prefer connecting nodes with dissimilar features. This method only works on graph with node features. For example it cannot work on Polblogs because the node features are unavailable.

- **GCNSVD** (Entezari et al., 2020): is a graph defense method based on GCN, which focuses on defending nettack Zügner et al. (2018). Since nettack is a high-rank attack, GCN-SVD pre-processes the perturbed graph with its low-rank approximation. It is straightforward to extend it to non-targeted and random attacks.

- **SimpleGCN** (Jin et al., 2021): utilizes similarity preserving aggregation to integrate the graph structure and the node features, and employs self-supervised learning to capture the similarity between node features. Notice SimpleGCN is not specifically designed for graph defense, and we find it also has good performance under the poisoning attacks, thus we add this method as another graph defense baseline.

## D.2 Description of Graph Attack Methods

Generally speaking, there are two types of the adversarial attacks on node classification tasks: test-time attack (evasion) and train-time attack (poisoning). In both types of attacks we first generate a perturbed adjacency matrix based on a victim model, and then in evasion attacks we test it directly on the victim model, and in poisoning attacks we train a new model with the perturbed adjacency matrix. For generating the adversarial perturbations, we apply three methods:

- **DICE** (Waniek et al., 2018): is a baseline attack method (delete internally, connect externally). In each perturbation, we randomly choose whether to insert or remove an edge. Edges are only removed between nodes from the same classes, and only inserted between nodes from different classes.

- **PGD** (Xu et al., 2019): calculates the gradient of the adjacency matrix, and the gradient serves as a probabilistic vector, then a random sampling is applied for generating a near-optimal binary perturbation based on this vector.

- **Metattack** (Zügner and Günnemann, 2019): was proposed to generate poisoning attacks based on meta-learning. It has an approximate version A-Metattack. In our experiments, we apply the original Metattack.

## D.3 Datasets Statistics

Cora and Citeseer (Sen et al., 2008) are citation datasets commonly used for evaluating GNNs. Polblogs (Adamic and Glance, 2005) is another common benchmark dataset where each node is a political blog. In Table 9 we provide the statistics for each graph. We preprocess the graph and only use the largest connected component.

Table 9: Dataset Statistics

| Datasets | Cora | Citeseer | Polblogs |
|---|---|---|---|
| #Nodes | 2708 | 3327 | 1222 |
| #Edges | 5429 | 4732 | 16714 |
| #Features | 1433 | 3703 | N/A |
| #Classes | 7 | 6 | 2 |

## D.4 Training Setup

**Optimization hyperparameters.** We use the Adam optimizer with a learning rate 0.01 and weight decay of 0.0005. All models are trained for 200 epochs with no weight scheduling. We add a dropout layer with rate $p = 0.5$ after each GNN layer during training. We apply no early stopping and the optimal model is selected with its performance on the validation set. The test set is never touched during training.

**Train/val/test split.** The evaluation procedures of GNNs on node classification tasks have suffered overfitting bias from using a single train-test split. Shchur et al. (2018) showed that different splits could significantly affect the performance and ranking of models. In all our experiments on node classification tasks, we apply the split setting in Zügner and Günnemann (2019), which utilizes 10% samples for training, 10% samples for validating, and 80% samples for testing. We generate 20 random splits and for each split we train 10 models with different random initialization. We report the mean and standard deviation of the accuracy of the 200 random models in our results.

### D.5 SETTINGS OF THE AVERAGE OF GRADIENT NORM

For the results in Sec. 5.2 we generate noise $z_A, z_X$ from Gaussian distribution $\mathcal{N}(A, \sigma^2 I)$ and $\mathcal{N}(X, \sigma^2 I)$, then calculate the $l_2$ norm of the loss gradient $||\nabla_A L_{\text{train}}(\theta; A, X)|_{A=z_A}||_2$ and $||\nabla_X L_{\text{train}}(\theta; A, X)|_{A=z_A}||_2$. In our experiments we choose $\sigma = 0.0005$, because we expect the perturbed input to be close to the clean input.

### D.6 SETTINGS OF VISUALIZATION OF LOSS LANDSCAPE

For the results in Sec. 5.3, we generate a random direction $u$ from a Gaussian distribution and perform $l_2$ normalization, it is equal to randomly selecting a direction on the $l_2$ unit ball. As seen in Fig. 9(a), there is a large gap between the final loss value of WT-AWP and vanilla GCN, we have to parallel move the loss landscape of WT-AWP and GCN to the same level for making comparison. The experiments are performed on Cora, similar results also hold for other datasets.

### D.7 HYPERPARAMETERS $(\lambda, \rho)$ FOR POISONING ATTACKS

Table 10: Hyperparameters of WT-AWP for poisoning PGD attack and Metattack of Sec. 5.4

| $(\lambda, \rho)$ | Cora | Citeseer | Polblogs |
|---|---|---|---|
| GCN | | | |
| GCNJaccard | (0.7, 0.5) | (0.7, 2) | (0.5, 0.5) |
| GCNSVD | | | |
| SimPGCN | (0.3, 0.5) | (0.5, 0.1) | (0.3, 2) Metattack (0.5, 0.5) PGD |

### D.8 RANDOMIZED SMOOTHING

Following Bojchevski et al. (2020), we create smoothed versions of our GNN models by randomly perturbing the adjacency matrix (or the node features) and predicting the majority vote for the randomly-perturbed samples. We denote with $p_a$ the probability of flipping an entry from 0 to 1, i.e. adding an edge or a feature, and with $p_d$ the probability of flipping an entry from 1 to 0, *i.e.* deleting an edge or a feature. In all experiments, for the certification of node features we generate random perturbations with $p_a = 0.01, p_d = 0.6$, and for perturbing the adjacency matrix we use $p_a = 0.001, p_d = 0.4$. We consider the prediction of the smoothed GNN for a given node correct if and only if it is correct and certifiably robust. This means the prediction of the node does not change for any perturbation within the radius (i.e. for any $r_d$ deletions or $r_a$ additions).

### D.9 GRAPH CLASSIFICATION

Table 11: Dataset Statistics.

| Datasets | Proteins | IMDB-B | IMDB-M |
|---|---|---|---|
| #Nodes (max) | 620 | 136 | 89 |
| #Nodes (avg) | 39.06 | 19.77 | 13.00 |
| #Graphs | 1113 | 1000 | 1500 |
| #Classes | 2 | 2 | 3 |

**Datasets.** We use three popular graph classification datasets, including one bioinformatics dataset Proteins, and two social network datasets IMDB-Binary and IMDB-Multi (Yanardag and Vishwanathan, 2015) for evaluation. The details are shown in Table 11.

**Settings.** We use 80% samples for training, 10% samples for validating and the rest 10% for testing. The baseline model is a two-layer GCN with 16 hidden dimension and a global mean pooling layer after the second graph convolution layer. A linear read-out layer is attached to the output of the GCN to generate predictions. We apply the same training settings for GCN and GCN+WT-AWP. We train both models for 200 epoches with the Adam optimizer, learning rate 0.01 and weight decay 0.0005. The best model is selected with only the validation accuracy. For each of GCN and GCN+WT-AWP we take 10 random initialization and report the average accuracy and standard deviation. The hyperparameters $(\lambda, \rho)$ of WT-AWP is (0.3, 0.5) for Proteins, (0.05, 0.1) for IMDB-M and (0.5, 0.1) for IMDB-B.

## D.10 NORM OF GRADIENT DURING TRAINING

In this experiment we train a vanilla GCN, GCN+AWP with $\rho = 0.1$, GCN+WT-AWP with $\lambda = 0.5, \rho = 1$ on Cora and plot the relative gradient norm $||\nabla\boldsymbol{\theta}||_2/||\boldsymbol{\theta}||_2$ during training. Both AWP and WT-AWP have small relative gradient norm compared to GCN when epoch is larger than 100.

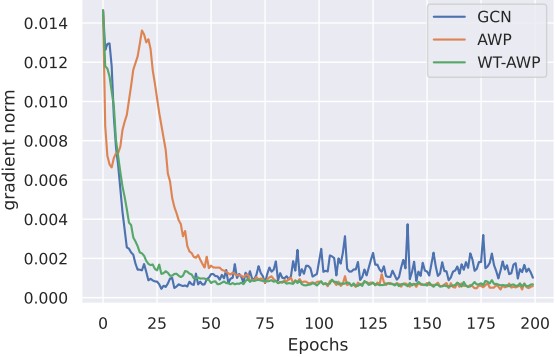

Figure 11: Norm of Gradient during training.

## D.11 ABLATION STUDY OF THE PERTURBED LAYER

Since in all experiment above we only perturb the first layer of GNN with WT-AWP, we provide experimental results corresponds to perturb only the second layer with WT-AWP. The backbone is GCN and the benchmark is Cora. As Table 12 shows, skipping the first layer in WT-AWP methods have worse performance than skipping the last layer.

Table 12: Ablation study with $\lambda$ and $\rho$ on WT-AWP, where we only use AWP on the last layer. The backbone model is GCN and the benchmark is Cora.

| WT-AWP (last layer) | $\rho = 0.05$ | $\rho = 0.1$ | $\rho = 0.5$ | $\rho = 1$ | $\rho = 2.5$ | $\rho = 5$ |
|---|---|---|---|---|---|---|
| $\lambda = 0.1$ | $84.09 \pm 0.62$ | $84.13 \pm 0.60$ | $84.18 \pm 0.60$ | $83.87 \pm 0.85$ | $81.40 \pm 1.92$ | $65.34 \pm 6.41$ |
| $\lambda = 0.3$ | $84.14 \pm 0.58$ | $84.12 \pm 0.64$ | $84.14 \pm 0.69$ | $82.30 \pm 1.47$ | $33.50 \pm 1.48$ | $29.18 \pm 0.00$ |
| $\lambda = 0.5$ | $84.13 \pm 0.60$ | $84.10 \pm 0.63$ | $84.08 \pm 0.77$ | $78.00 \pm 3.16$ | $29.18 \pm 0.00$ | $29.18 \pm 0.00$ |
| $\lambda = 0.7$ | $84.12 \pm 0.62$ | $84.14 \pm 0.64$ | $83.74 \pm 0.84$ | $67.37 \pm 5.01$ | $29.18 \pm 0.00$ | $29.18 \pm 0.00$ |
| $\lambda = 1.0$ | $84.20 \pm 0.62$ | $84.19 \pm 0.65$ | $82.80 \pm 1.04$ | $29.18 \pm 0.02$ | $29.18 \pm 0.00$ | $29.18 \pm 0.00$ |

# E GENERALIZATION BOUND ON GNN NODE CLASSIFICATION

**Theorem 3.** *(generalization bound) Assuming $L_{all}(\theta; A, X) \leq \mathbb{E}_{z \sim \mathcal{N}(0, \Sigma)}[L_{all}(\theta + z; A, X)]$, for any set of training nodes $\mathcal{V}_{train}$ from $\mathcal{V}_{all}$, $\forall m > \sqrt{d}$, with probability at least $1 - \delta$, we have*

$$L_{all}(\theta; A, X) \leq \max_{||\delta||_2 \leq \rho} [L_{train}(\theta + \delta; A, X)] + (\frac{m^2}{d} e^{1 - \frac{m^2}{d}})^{d/2}$$
$$+ \frac{1}{\sqrt{N_0}} \left( \frac{1}{2} \left[ 1 + d \log(1 + \frac{m^2 ||\theta||_2^2}{d\rho^2}) \right] + \ln \frac{3}{\delta} + \frac{1}{4} + \Theta(K \cdot \epsilon_{all}) \right). \tag{12}$$

*where $d$ is the number of parameters in the GNN, $K$ is the number of groundtruth labels, $\epsilon_{all}$ is a fixed constant w.r.t. $\mathcal{V}_{all}$, $N_0$ is the volume of $\mathcal{V}_{train}$.*

We use the assumption in Foret et al. (2021), $L_{all}(\theta; A, X) \leq \mathbb{E}_{z \sim \mathcal{N}(0, \Sigma)}[L_{all}(\theta + z; A, X)]$, which means that adding Gaussian perturbation should not decrease the test error.

*Proof.* Our proof is motivated by the subgroup generation bound on node classification tasks (Ma et al., 2021) and the intuition of theorem 1 in Foret et al. (2021).

**Lemma 1.** *(PAC-Bayes bound node classification tasks (Ma et al., 2021)) For any set of training nodes $\mathcal{V}_{train}$ from $\mathcal{V}_{all}$, for any subgroup of nodes $\mathcal{V}_m \subset \mathcal{V}_{all}$, for any prior distribution $\mathcal{P}$, with probability at least $1 - \delta$, for any distribution $\mathcal{Q}$ we have*

$$\mathbb{E}_{\theta \sim \mathcal{Q}}[L_m(\theta; A, X)] \leq \mathbb{E}_{\theta \sim \mathcal{Q}}[L_{train}(\theta; A, X)] + \frac{1}{\sqrt{N_0}} (\mathbb{D}_{KL}(\mathcal{Q}||\mathcal{P}) + \ln \frac{3}{\delta} + \frac{1}{4} + \Theta(K \epsilon_m)) \tag{13}$$

*where $N_0$ is the volume of the training set $\mathcal{V}_{train}$, $K$ is the total number of classes, $\epsilon_m$ is a constant depend on the subgroup $\mathcal{V}_m$.*

If we take $\mathcal{V}_m = \mathcal{V}_{all}$ in Lemma 1, we have

$$\mathbb{E}_{\theta \sim \mathcal{Q}}[L_{all}(\theta; A, X)] \leq \mathbb{E}_{\theta \sim \mathcal{Q}}[L_{train}(\theta; A, X)] + \frac{1}{\sqrt{N_0}} (\mathbb{D}_{KL}(\mathcal{Q}||\mathcal{P}) + \ln \frac{3}{\delta} + \frac{1}{4} + \Theta(K \epsilon_{all})) \tag{14}$$

Assume both $\mathcal{P}$ and $\mathcal{Q}$ are Gaussian distributions with diagonal covariance matrix, i.e. $\mathcal{P}_i \sim \mathcal{N}(\mu_p, \sigma_p^2 I_d)$, $\mathcal{Q}_i \sim \mathcal{N}(\boldsymbol{\mu}_q, \sigma_q^2 I_d)$, where $d$ is the dimension of $\theta$, we have

$$\mathbb{D}_{KL}(\mathcal{Q}||\mathcal{P}) = \frac{1}{2} \left[ d \log \frac{\sigma_p^2}{\sigma_q^2} - d + d\frac{\sigma_q^2}{\sigma_p^2} + ||\frac{\mu_p - \mu_q}{\sigma_p}||_2^2 \right], \tag{15}$$

Take $\mu_q = \theta, \mu_p = 0$, we expect the KL-divergence $\mathbb{D}_{KL}(\mathcal{Q}||\mathcal{P})$ to be as small as possible w.r.t. $\sigma_p$.

**Lemma 2.** *(Foret et al., 2021) Take $\mu_q = \theta, \mu_p = 0$. There exist pre-defined $\sigma_p$ such that*

$$\mathbb{D}_{KL}(\mathcal{Q}||\mathcal{P}) \leq \frac{1}{2} \left[ 1 + d \log(1 + \frac{||\theta||_2^2}{d\sigma_q^2}) \right] \tag{16}$$

Thus we have the generalization bound

$$\mathbb{E}_{\theta \sim \mathcal{Q}}[L_{all}(\theta; A, X)] \leq \mathbb{E}_{\theta \sim \mathcal{Q}}[L_{train}(\theta; A, X)]$$
$$+ \frac{1}{\sqrt{N_0}} \left( \frac{1}{2} \left[ 1 + d \log(1 + \frac{||\theta||_2^2}{d\sigma_q^2}) \right] + \ln \frac{3}{\delta} + \frac{1}{4} + \Theta(K \epsilon_{all}) \right) \tag{17}$$

As $\mathcal{Q} \sim \mathcal{N}(\theta, diag\{\sigma_q^2\})$, consider $z \sim \mathcal{N}(0, diag\{\sigma_q^2\})$, we have $\frac{z}{\sigma_q} \sim \mathcal{N}(0, I_d)$ and $\mathbb{E}_{\theta \sim \mathcal{Q}}[L_{\text{train}}(\theta; A, X)] = \mathbb{E}_z[L_{\text{train}}(\theta + z; A, X)]$. Thus we have $\forall m > 0$

$$
\begin{aligned}
&\mathbb{E}_{\theta \sim \mathcal{Q}}[L_{\text{train}}(\theta; A, X)] \\
&= \mathbb{E}_z[L_{\text{train}}(\theta + z; A, X)] \\
&= \mathbb{E}_z[L_{\text{train}}(\theta + z; A, X)| \, ||\frac{z}{\sigma_q}||_2 \leq m]\mathbb{P}(||\frac{z}{\sigma_q}||_2 \leq m) \\
&+ \mathbb{E}_z[L_{\text{train}}(\theta + z; A, X)| \, ||\frac{z}{\sigma_q}||_2 > m]\mathbb{P}(||\frac{z}{\sigma_q}||_2 > m)) \\
&\leq \max_{||\delta||_2 \leq m\sigma_q}[L_{\text{train}}(\theta + \delta; A, X)]\mathbb{P}(||\frac{z}{\sigma_q}||_2 \leq m) + \mathbb{P}(||\frac{z}{\sigma_q}||_2 > m). \\
&\leq \max_{||\delta||_2 \leq m\sigma_q}[L_{\text{train}}(\theta + \delta; A, X)] + \mathbb{P}(||\frac{z}{\sigma_q}||_2 > m).
\end{aligned}
\tag{18}
$$

As $\frac{z}{\sigma_q} \sim \mathcal{N}(0, I_d)$, by Chernoff bound of chi-squred distribution we have when $m > \sqrt{d}$,

$$
\mathbb{P}(||\frac{z}{\sigma_q}||_2 > m) \leq (\frac{m^2}{d}e^{1-\frac{m^2}{d}})^{d/2}
\tag{19}
$$

Thus

$$
\mathbb{E}_{\theta \sim \mathcal{Q}}[L_{\text{train}}(\theta; A, X)] \leq \max_{||\delta||_2 \leq m\sigma_q}[L_{\text{train}}(\theta + \delta; A, X)] + (\frac{m^2}{d}e^{1-\frac{m^2}{d}})^{d/2},
\tag{20}
$$

combining it with Eq. 17 and denote $\sigma_q = \rho/m$ we have

$$
\begin{aligned}
L_{\text{all}}(\theta; A, X) \leq \max_{||\delta||_2 \leq \rho}[L_{\text{train}}(\theta + \delta; A, X)] + (\frac{m^2}{d}e^{1-\frac{m^2}{d}})^{d/2} \\
+ \frac{1}{\sqrt{N_0}}\left(\frac{1}{2}\left[1 + d\log(1 + \frac{m^2||\theta||_2^2}{d\rho^2})\right] + \ln\frac{3}{\delta} + \frac{1}{4} + \Theta(K\epsilon_{all})\right).
\end{aligned}
\tag{21}
$$

$\square$

