# OpenReview forum: "Adversarial Weight Perturbation Improves Generalization in Graph Neural Networks"
_ICLR.cc/2022/Conference — ICLR 2022 Submitted_

### Official Review · Reviewer_gQzb · 2021-10-25

**Correctness:** 3
**Technical Novelty And Significance:** 2
**Empirical Novelty And Significance:** 2
**Recommendation:** 3
**Confidence:** 4

**Main Review:**

Strengths: I am satisfied with the numerical experiments in section 5, although they largely follows the routes of existing work.

Weaknesses: I will point the weaknesses by sections.

Section 3: i) The theoretical results are very incremental based on existing works e.g., Wu et al., 2020a, which can be easily derived or directly used. ii) the AWP algorithm listed here is the same as Wu et al., 2020a (only add a letter A).

Section 4: i) After I read the paper thoroughly twice, I do not see how you assign $\theta^{(awp)}$ and $\theta^{(normal)}$, which is very important. If you have that in your paper, first let me know where I can find it, second move that to section 4 and explain it. It is unacceptable not having it in section 4. ii) Do not have theory or even intuition about T-AWP and W-AWP. In my perspective, how you justify your algorithms is very important, which is much important than the so called "theoretical" results in Section 3. You can even save the space in section 3 for you to defend your algorithms in section 4. I believe what I point out here should be your novelty for this paper. At least, please, add some good intuitions for your algorithms (I will add points if you do that.) iii) Please add the WT-AWG result in Figure 3.

Summary: Good: experiments; Weak: i) too incremental and lack of novelty; ii) do not provides theoretical/intuitive explanation of the proposed algorithms.







**Summary Of The Paper:**

This work studies how AWP improves the GNN, mainly at the generalization aspect. The authors first derive several theoretical results that can be easily derived from the existing works on CNN, e.g., Wu et al., 2020a and Foret et a;., 2021 in section 3.. In section 4, the authors show that if they directly apply large $\rho$ as suggested by (3), they will encounter the gradient vanishing phenomenon during the training. Based on this phenomenon, the authors propose two ways to solve this. One is truncated AWP, the other is weighted AWP. Then, the authors conduct simple experiments using Vanilla GCN, GCN+AWP, GCN+TAWP, GCN+WAWP, where Vanilla perform the best whereas GCN+TAWP, GCN+WAWP have smoother decision boundary. And they decide to combine T+W with AWP (to be frank, I do not know why they do not add experiment result of WT-AWP here.) In section 5, the authors conduct a lot of experiments like clean accuracy, robust accuracy, etc, which follow the existing works.

The contribution as I can see is that the authors are the first ones to conduct extensive experiments using AWP on graph dataset. However, I do not see other contributions listed by the authors, which I will explain later in the main review.

**Summary Of The Review:**

This is work i) is quite incremental compared to the existing works, ii) does not explain/justify/describe the algorithms well. I do not think this paper is good enough for ICLR.

---

> ### Author Response · Authors · 2021-11-19
> **Author Response to Reviewer gQzb**
>
> We thank the reviewer for the valuable review. In the following we address each of the raised points individually.
>
> *Theoretical contribution.*
>
> In the updated version of the paper we derive a new generalization bound that applies to tasks with non-i.i.d. data such as node classification. See the reply to all reviewers for a detailed discussion of the new theorem. However, we would like to point out that theory is not the main focus of this paper, and the rationale for including this result is to provide a justification for why the proposed method should work. Our main goal was to empirically evaluate AWP for GNNs. We would also like to point out that the algorithm in Wu et al., 2020a does not consider weighting or truncation, both of which are important to prevent the vanishing gradient issue.
>
> *How do we assign $\theta^{awp}$*
>
> We have indicated the perturbed layer $\theta^{awp}$ in the footnote of the page 5, where we only use AWP on the first layer, "perturb" means we apply AWP to this layer. In section 5, we explain the perturbed layer in the second sentence of paragraph "Settings for WT-AWP".
>
> *Intuition of algorithm and missing WT-AWP plot.*
>
> We add the intuition for our WT-AWP method in Sec. 4 and the missing WT-AWP result in Figure 3. The intuition for our algorithm is as follows: The vanishing gradient is mainly due to the exploding of the logit values, which is caused by perturbing all layers in the model. Thus, a natural idea is to only apply AWP on certain layers in order to mitigate the logit explosion, and this it the truncated AWP. Another idea is to provide a second source of valid gradients to the model which we do by adding the the vanilla loss $L_\text{train}(\theta;A,X)$ to the AWP objective. Even when the AWP loss suffers from the vanishing gradient issue, the vanilla loss will not be affected. In addition, the newly added generalization bound provides a justification for the general AWP approach.

---

> > ### Comment · Reviewer_gQzb · 2021-11-19
> > **Updated Review**
> >
> > After I read the response from the authors, I decide to maintain my evaluation as before. Main reasons are as follows:
> >
> > i) As said by the authors in the response, the main focus of this paper is "empirically evaluate AWP for GNNs". Then the only difference between this work and previous work using AWP is just by adding $\hat{A}$ matrix before the $XW$ in each layer. So, it is quite an incremental work.
> >
> > ii) Meanwhile, I do not see good justification of the proposed algorithms, and even a proper description of the proposed algorithms (like how to assign $\theta^{awp}$).

---

> > > ### Author Response · Authors · 2021-11-21
> > > **Response to the Updated Review**
> > >
> > > Thanks for your updates
> > >
> > > *1. Our contributions*
> > >
> > > We respectively disagree that "the **only** difference between this work and previous work using AWP is just by adding $\hat{A}$ matrix before the $XW$ in each layer". In our work we found the vanishing gradient problem in the regular AWP methods and propose an improved method WT-AWP which alleviates this issue. In our ablation study (Sec 5.7, Table 4), we show that blindly applying AWP to GNNs leads to negligible improvement.
> > >
> > > *2. Description of our algorithm*
> > >
> > > In the WT-AWP algorithm we apply a numerical optimizer such as Adam to the WT-AWP objective
> > >
> > > $L_{WT-AWP}(\theta) = [\lambda L_\text{train}({{\theta}}+[\hat{{\delta}}^{(awp)*}({{\theta}}^{(awp)}),0];{A},{X})+(1-\lambda)L_\text{train}({{\theta}};{A},{X})]$
> > >
> > > Since in our empirical experiments the GNNs always have a 2-layer structure, we assign $\theta^{awp}$ as the first layer and $\theta^{normal}$ as the last layer.
> > > Then the perturbation $\hat{{\delta}}^{(awp)*}$ (first layer) is computed via Eq. (4). Next the gradient $g$ of $L_{WT-AWP}$ is calculated with Eq. (6).
> > >
> > > $g = \lambda\nabla_{\theta} L_\text{train}(\theta;A,X)\vert_{\theta_{t-1}+[\hat{\delta}^*(\theta_{t-1}^{(awp)}),0]}+(1-\lambda)\nabla_{\theta} L_\text{train}(\theta;A,X)\vert_{\theta_{t-1}}$
> > >
> > > Finally we update the weight via $\theta_t = \theta_{t-1}-\alpha g$.
> > >
> > > We will add this description in our updated paper (see Appendix B.1).
> > >
> > > *3. Assignment of $\theta^{awp}$ and $\theta^{normal}$*
> > >
> > > Since in all experiments we use two-layer GNNs, we assign $\theta^{awp}$ = first layer, and $\theta^{normal}$ = last layer. We will mention this explicitly in our paper.

---

### Official Review · Reviewer_MYVq · 2021-10-28

**Correctness:** 4
**Technical Novelty And Significance:** 2
**Empirical Novelty And Significance:** 2
**Recommendation:** 5
**Confidence:** 3

**Main Review:**

**Strengths:**
The line of work concerning adversarial weight perturbations is interesting and significant, as it is one of a few locations where the nascent theory of deep networks can provide easy tweaks to training that improve generalization. Identifying a problem with the current approach (even on MLPs) is valuable and significant, and ample evidence is presented to suggest the vanishing gradient problem is actually what is occurring. The proposed solutions are simple and easy to implement. The experimentation is thorough: while the improvements provided by AWP are very minor (less than a percent in some instances), it is yet another easy and cheap trick to ever-so-slightly boost the performance of a network.

**Weaknesses:**
First, while it is an important insight to notice the vanishing gradient, the proposed remedies (weight truncation and weighted AWP) are natural and not particularly novel. These are likely the first things that one would try to mitigate the observed problems with AWP and don't represent a great stride forward in the field. Similarly, it is not particularly surprising that the benefits of AWP as evidenced in MLPs carry over to GNNs: not enough motivation or evidence is presented to make this seem surprising or unexpected. Third, Theorem 1 is trivial. If the definition of Ltrain is to be the one that only performs a single first-order step, then of course the gradient evaluated at a minimum is going to be zero: the interesting question here is how the true AWP loss (eq2) relates to the standard training loss. This theorem doesn't add add anything to the story.

**Questions:**
- One of my complaints is about the novelty of contribution re: GNNs vs MLPs. Did I miss this, or is there simply a much more pronounced effect of the gradient-vanishing phenomenon in GNNS than MLPs?
- How much do the gradient norms (with respect to the weights) actually change when training under AWP vs WT-AWP vs standard training?

**Summary Of The Paper:**

The authors extend the line of work related to adversarial weight perturbations: first showing that a vanishing gradient issue in standard AWP can hinder training. To remedy this, some natural tweaks are applied to AWP. The authors then focus on using AWP to train graph neural networks, demonstrating a minor boost in both clean and robust accuracy.


**Summary Of The Review:**

This is an interesting line of work, and the pointing out of (and subsequent fixing of) the gradient-vanishing phenomenon in AWP is a valuable contribution. Past this, none of the results are strikingly novel or groundbreaking. I think this is a borderline paper, tending towards rejection, but could be convinced to boost my score slightly if I've misunderstood something.

---

> ### Author Response · Authors · 2021-11-19
> **Author Response to Reviewer MYVq**
>
> We thank the reviewer for the valuable review. In the following we address each of the raised points individually.
>
> *Novelty and contribution of our work.*
>
> Our main goal was to investigate AWP for GNNs. Our main contribution is to empirically show that WT-AWP can improve the performance and robustness of GNNs on various node classification tasks. In our ablation study (Sec. 5.7) we show that blindly applying AWP to GNNs will not lead to a significant improvement in the performance. We agree that the proposed remedies are natural, and moreover they are also simple and easy to implement. We believe that this is a strength and not a weakness, especially since they consistently boost the performance and robustness. While the results may not be surprising to some readers we believe that our thorough empirical validation is a valuable contribution in itself.
>
> In addition, in the updated version of the paper we derive a new generalization bound that applies to tasks with non-i.i.d. data such as node classification. See the reply to all reviewers for a detailed discussion of the new theorem. However, we would like to point out that theory is not the main focus of this paper, and the rationale for including this result is to provide a justification for why the proposed method should work.
>
> *Theorem 1 and the "true AWP" (Eq. 2).*
>
> Since the AWP loss used for training is closer to Eq. 5, we think that it is more valuable from a practical point of view to analyze Eq. 5 rather than the "true AWP" in Eq. 2. Even though Theorem 1 (Theorem 2 in the updated version) is technically quite simple, we think that it still provides us with useful insights.
>
>
> *Vanishing gradient regarding GNNs vs. MLPs.*
>
> The gradient-vanishing phenomenon due to adversarial weight perturbation of either GNNs or MLPs has not been studied before, and as far as we know we are the first to raise this issue for both. In earlier work (Stutz et al. and Wu et al.) the neural networks they study are deep and the vanishing gradient problem discussed is thus even worse. This is the reason why they are forced to use a small perturbation strength $\rho$, even though larger values of $\rho$ can lead to a tighter bound and find flatter minima (see also the reply to reviewer 2WEe on why a large $\rho$ is useful). Note, our proposed WT-AWP solution could also be applied to i.i.d. data such as studied in earlier works, but this is not our focus.
>
>
> *Gradient norm during training*
>
> We plot the gradient norm during training in Appendix D.10. In this experiment we train a vanilla GCN, GCN+AWP with $\rho=0.1$, GCN+WT-AWP with $\lambda = 0.5,\rho=1$ on Cora and we plot the relative gradient norm $||\nabla\theta||_2/||\theta||_2$ during training. Both AWP and WT-AWP have small relative gradient norm compared to GCN when the epoch is larger than 100. However, as reviewer 2WEe also points out, the gradient norm may not be the best indicator of flatness.

---

### Official Review · Reviewer_2WEe · 2021-11-02

**Correctness:** 3
**Technical Novelty And Significance:** 2
**Empirical Novelty And Significance:** 3
**Recommendation:** 6
**Confidence:** 4

**Main Review:**

Strengths:
- The paper is generally clearly structured and written; the toy example visualizations are nice and the algorithm helps understanding.
- An interesting problem/question is addressed: how to effectively use AWP/sharpness-aware minimization for graph neural networks.
- It is also nice to show that the optimum does not change with AWP training.
- Experiments for clean and robust accuracy with several models are conducted. Besides mean, standard deviation is reported which is nice to judge the improvement.
- The approach improves across the board, even though improvement is small at times.
- Analysis in terms of flatness (gradients and visualization) and ablation studies are provided.

Weaknesses:
- In the introduction, the authors state the i.i.d assumption as an important criteria for existing flatness approaches. However, it is not discussed when this becomes relevant. As far as I am aware, the i.i.id assumption does not play an important role in related discussion/papers. Could the authors comment on that? As I understand the paper, the main point is to see how these approaches work in non i.i.d settings such as node classification, but the paper does not really theoretically discuss this issue.
- One paper that could be included in related work as it is quite related:

[a] https://arxiv.org/abs/1609.04836

- Also a discussion of scale-invariance and the criticism of [b] is missing (this should be handled as in Stutz et al., but it would be interesting at least to say why this generalizes to the GCNs considered)

[b] https://arxiv.org/pdf/1703.04933.pdf

- Notation-wise, the main sections could be improved by making explicit that a per-layer neighborhood B is used from the beginning. Currently, this is made clear in Eq. (4) while Eq. (1) to (3) suggest that all parameters are appended.
- Also, are the GCN baselines equipped with biases in addition to the weights? If so, are biases and weights treated as separate layers (as in Stutz et al.)?
- In terms of contributions, the theoretical contributions (mainly Thm 1) and methodological contributions seem a bit limited. Although I have not seen Thm 1 in other papers (which makes it refreshing to actually see it), I found that the result is very intuitive and the proof is also quite straight-forward. The techniques employed to improve AWP also seem very specific to graph problems where larger \rho are used than for vision problems. Thus, I see the contributions mostly in verifying this approach on graph data.
- Regarding the vanishing gradient problem: I am having difficulties understanding why \rho has to be chosen as large as done throughout the toy example and the experiments. Stutz et al. And Wu et al. Consider very small \rho of 0.5% (=0.005 or lower). Obviously these are much deeper networks and not graph neural networks. I am wondering if the authors could give more details on why a large \rho is needed. Is it because of the 2-layer structure or the architecture differences?
- The two proposed approaches do actually not improve performance on the toy example. While I understand that it is meant for illustration purposes, I believe it is badly chosen. I see that without truncation and weighting the accuracy is very bad, but shouldn’t you show that you can improve over the baseline of 98% and not be stuck at 95%? Did you try optimizing hyper-parameters, or is it a problem where TW-AWP just does not help (which would be interesting!)?
- Regarding 5.2, I am not entirely convinced that the gradient norm is the best indicator of flatness exactly because of the scale-invariance argumentation of Stutz et al.: Can the authors comment on that? I guess that the used GCNs do not use batch normalization (I have not seen BN used for graph neural networks before), but scale-invariance is a problem as described in [b]. Also, Fig. 4 (c) and (d) has some outliers that are not really explained.
- Table 2 is hard to read and very small.
- Some ablation that I find missing: ablation regarding layers. Which layer to skip? Obviously, the networks are two-layer networks, but I would find it very interesting whether it is always the last layer to skip (also in deeper networks) or always the first layer to perturb (also in deeper networks).

**Summary Of The Paper:**

The authors propose a variant of adversarial weight perturbations / sharpness aware minimization for graph (convolutional) neural networks for node and graph classification. In particular, they make two adjustments: “truncating”, i.e., limiting the weight perturbation to specific layers, and weighting the sharpness aware loss with the regular loss during training.

**Summary Of The Review:**

I appreciate this paper in terms of applying AWP to graph neural networks and showing how it needs to be adapted to work well. Methodological contributions are small, however, and improvements vary across datasets and cases.

---

> ### Author Response · Authors · 2021-11-19
> **Author Response to Reviewer 2WEe**
>
> We thank the reviewer for the detailed and constructive review. In the following we address each of the raised points individually.
>
> *The i.i.d. assumption.*
>
> Thank you for the valuable suggestion. The i.i.d. assumption is important for existing flatness approaches, because the standard PAC-Bayes bound that they rely on only holds for i.i.d. data. Since this assumption does not hold for node classification we derived a new non-i.i.d. generalization bound. See the reply to all reviewers for a detailed discussion of the theorem added in the updated version.
>
> *Discussion of [a] and [b].*
>
> We now include a discussion of the suggested related works [a] and [b] in the updated paper. Thank you for the suggestion.
> The observation in [a] shows that large-batch training may reach sharp minima with poor generalization is relevant, however in GNNs the batch size is usually small. The scale-invariance issue raised in [b] may also affect GNNs, however we leave the investigation of this issue for future work.
>
>
>
> *Notation about  per-layer neighborhood B.*
>
> We agree with you and we improved the notation of Eq. 1 and Eq. 3 in the updated version.
>
> *The bias term in GNNs.*
>
> We keep the bias terms in the GNN models, but we do not perturb them in our AWP variants.
>
> *Why we need a large $\rho$?*
>
> There are two reasons. (1) Increasing $\rho$ can make the bound we derived in Theorem 1 tighter since the additional term $h()$ in Eq. 3 is a monotonically increasing function of $\rho$. (2) AWP helps to find flat local minima by skipping over sharp minima. Increasing $\rho$, increases the ammount of sharpening to the the local minima. Thus, we expect a moderately large $\rho$ will skip the sharp minima during training. Stutz et al. and Wu et al. select small $\rho$ out of necessity. Since the neural networks they study are deep, the vanishing gradient problem discussed in our work is exacerbated and larger values of $\rho$ make training more difficult. Note, our proposed solutions could also be applied to i.i.d. data such as studied in those works, but this is not our focus.
>
> *Poor accuracy of W-AWP and T-AWP on the 2D toy dataset.*
>
> You are right, we only use the toy example to show the vanishing gradient problem. As our dataset is only 2D and linearly separable, the GCN easily generalizes to the test set without suffering from problems such as overfitting. Thus, in this case W-AWP and T-AWP cannot help to improve the results since they are already quite good. Besides, in the original plots we did not optimize the hyperparameters of W-AWP and T-AWP. After some hyperparameter tuning, W-AWP and T-AWP achieve similar accuracy as the vanilla model (see Fig. 3 in the updated paper).
>
> *Gradient norm in 5.2.*
>
> Our GNNs do not use a batch normalization layer so it is not possible to scale the weights as in Stutz et al.
> We agree that the gradient norm w.r.t. the *weights* is not the best indicator to show the flatness. However, in Sec. 5.2 we are calculating the gradient norm w.r.t. the *inputs* (i.e. the adjacency matrix and the features) rather than the weights. Thus, scaling is not an issue here.
> Despite the observation that sharp minima (w.r.t. weights) can also generalize for neural networks [b], models which are smooth w.r.t. the inputs are less sensitive to adversarial perturbations and thus have better generalization. The goal of this experiment was precisely to study this smoothness, i.e. the stability of the output under input perturbations. In almost all cases models trained with WT-AWP have smaller gradient norm (y-axis) compared to a vanilla GCN. The "outliers" are WT-AWP models with  worse clean accuracy (x-axis). Thus, they are not "outliers" from the flatness perspective.
>
> *Which layer to skip in WT-AWP?*
>
> In general we skip the last layer in all experiments, because empirically this resulted in a better performance. For completeness, in the updated version we added an ablation study w.r.t. skipping different layers (see Appendix D.11).

---

> > ### Comment · Reviewer_2WEe · 2021-11-28
> > **Thanks for the Answers**
> >
> > I thanks the reviewers for answers and comments on the raised points. All my questions were addressed appropriately.
> >
> > I decided to keep my score at 6.

---

### Official Review · Reviewer_DWF4 · 2021-11-02

**Correctness:** 2
**Technical Novelty And Significance:** 2
**Empirical Novelty And Significance:** 3
**Recommendation:** 6
**Confidence:** 2

**Main Review:**

This paper focuses on extending AWP to GNN. This paper analyzes the vanishing-gradient issue existing in AWP and gives a more detailed theoretical proof. The experimental part of this article is comprehensive, and the experimental results also verify the advantages of the proposed method.

However, there still exist some insufficiency and confusing places.

1, What is the term h in equation(3). The statements said h is a monotonously increasing function. Where is it from?

2, The paper utilized comprehensive experiments to show the efficiency when facing attacks, but no analysis and theoretical proof for this advantage. I think the source of the advantage in handling attacks is worth clarification. Especially, in Table 2, some experimental results under attacks are even better than the natural setting.

3, Figure1 and Figure3 are blurred. I can not distinguish face color and border color. It is recommended to use vector graphics.

4, The sentence on Page 5 footnotes "Perturbing only the second layer instead performs similarly." is confusing for me.


**Summary Of The Paper:**

The paper designed and tested WT-AWP, a new adversarial weight perturbation approach, on graph neural networks. They demonstrated that by locating flat local minima, our WT-AWP can improve the regularization of GNNs. They carried out comprehensive tests to verify the method. WT-AWP reliably increases GNN performance on a wide range of graph learning tasks, including node classification, graph defense, and graph classification.

**Summary Of The Review:**

The paper is well-organized and has comprehensive experiments to defense its method. However, some statements and equations are not clear enough. The illustrations and figures are not meticulous. I hope the authors can further polish the paper in detail.

---

> ### Author Response · Authors · 2021-11-19
> **Author Response to Reviewer DWF4**
>
> We thank the reviewer for the thoughtful review and helpful comments. In the following we address each of the raised points individually.
>
> *1. Term h in Eq. 3.*
>
> In the Appendix E in the updated paper, we show the explicit form of $h()$ for our newly derived generalization bound. We see that $h()$ is a monotonically increasing function of $\theta$ that controls the tightness of the bound.
>
> *2.1 Advantage of WT-AWP against attacks.*
>
> In Section 5.2 we empirically show that models trained with (WT-)AWP have smaller averaged gradient norm w.r.t. the inputs, i.e. they are smoother w.r.t. the inputs. Intuitively, this implies that models trained with AWP are less sensitive to input perturbations, and thus less sensitive to adversarial attacks. While we do not have any theoretical results for the performance under attacks, our new generalization bounds (see the shared reply to all reviewers) provide some insight into why AWP can improve generalization.
>
>
> *2.2 Results under attacks are better than the natural settings.*
>
> We notice that this only happens with PGD attacks and graph defense methods. Because PGD attacks on graphs (unlike on images) are not strong, the improvement of WT-AWP is greater than the decrease in accuracy due to the attack. For Metattack, which is stronger, the improvement due to WT-AWP cannot overcome the reduction in accuracy due to the attack. Besides, as the graph defense methods focus on improving robustness rather than clean accuracy, it is possible that with the (not strong) PGD attacks and WT-AWP, the robust accuracy outperforms the clean accuracy.
>
> *3. Blurred figures.*
>
> Thank you for the suggestion. We updated the figures with vector graphics in the revision.
>
> *4. Confusing sentence on page 5.*
>
> Because in Truncated AWP, we only apply adversarial weight perturbations on some (but not all) layers, in the toy experiments we perturbed only the first layer for T-AWP objective. This goal of this sentences was to state that if we only perturb the second layer (instead of only the first) the performance was similar.

---

### Author Response · Authors · 2021-11-19
**Shared Reply to All Reviewers**

*New theoretical results.*

Since several reviewers raised a few related questions regarding the theoretical contributions in the paper, we address them jointly in this comment.

In the updated version of the paper we include an extended theoretical analysis that provides justification for the method we propose. Specifically, we leverage a recently proposed subgroup PAC-Bayes bound [1] that can be applied to tasks with non-i.i.d. data such as node classification.

The i.i.d. assumption is important for existing flatness/sharpness approaches [2, 3], because the standard PAC-Bayes bound [4] that they rely on holds only for i.i.d. data. Thus, the generalization bounds derived in previous work [2,3] cannot be directly applied to the semi-supervised node classification task which we study in our work. However, leveraging the results in [1], we were able to derive a new generalization bound. See the updated (informal) Theorem 1 in the revised main paper and the formal version and detailed proof in Appendix E. Note that in [1], the authors study standard training, while our bound extends to the adversarial weight perturbation setting. Even though the proof is not technically involved, the theory still provides useful insights. Namely, we see that we can bound the loss on all nodes by the perturbed loss on the training nodes, plus and additional term $h()$ that depends on the perturbation strength $\rho$. Increasing the perturbation strength makes the bound sharper but can lead to vanish gradient issues, justifying the need to fix them. In the Appendix E we also show the explicit form of $h()$ which is a monotonically increasing function of $\theta$ with additional constants (e.g. that depend on the training nodes and the number of parameters).

In the updated version of the paper we mark the changed parts with red for convenience.

References:

[1] Ma et al. "Subgroup Generalization and Fairness of Graph Neural Networks". NeurIPS 2021.

[2] Foret et al. Sharpness-aware Minimization for Efficiently Improving Generalization. ICLR 2021.

[3] Wu et al. Adversarial Weight Perturbation Helps Robust Generalization. NeurIPS 2020.

[4] Dziugaite et al. Computing Nonvacuous Generalization Bounds for Deep (Stochastic) Neural Networks with Many More Parameters Than Training Data. UAI 2017.

---

### Public Comment · ~Jean_Kaddour1 · 2022-03-10
**Thank you for this paper! We recently investigated SAM + GNNs.**

Thank you very much for writing this paper. I find it very interesting, especially the vanishing-gradient issue of AWP.

We have recently investigated SAM (and SWA) across 6 OGB-benchmarks and multiple GNN architectures: https://arxiv.org/pdf/2202.00661.pdf

I hope you find it complementary and relevant to your study.

Warm regards
Jean

---

### Decision · Program_Chairs · 2022-01-20

**Decision:**

Reject

**Comment:**

This paper considers a variant of adversarial weight perturbations / sharpness aware minimization for graph (convolutional) neural networks for node and graph classification. In particular, they make two adjustments: “truncating”, i.e., limiting the weight perturbation to specific layers, and weighting the sharpness aware loss with the regular loss during training. The reviewers found that the theoretical justifications (characterization of vanishing gradient and understanding of non-iid setting which was added during rebuttal) are interesting, but several reviewers also found the solution/empirical results not convincing enough. I recommend the authors to either shift the focus to the theoretical results or to strengthen the empirical results (and their connections with theory) following the comments of the reviewers.